# Yeast NDI1 reconfigures neuronal metabolism and prevents the unfolded protein response in mitochondrial complex I deficiency

**Lucy Granat**[1], **Debbra Y. Knorr**[1], **Daniel C. Ranson**[1], **Emma L. Hamer**[1], **Ram Prosad Chakrabarty**[2], **Francesca Mattedi**[1], **Laura Fort-Aznar**[3,4], **Frank Hirth**[1], **Sean T. Sweeney**[3], **Alessio Vagnoni**[1], **Navdeep S. Chandel**[2], **Joseph M. Bateman**[1]*

1 Maurice Wohl Clinical Neuroscience Institute, King's College London, London, United Kingdom, 2 Department of Medicine and Biochemistry & Molecular Genetics, Northwestern University Feinberg School of Medicine, Chicago, Illinois, United States of America, 3 Department of Biology and York Biomedical Research Institute, University of York, Heslington, York, United Kingdom, 4 Alzheimer's disease and other cognitive disorders Unit, Hospital Clínic de Barcelona IDIBAPS, Universitat de Barcelona, Barcelona, Spain

* joseph_matthew.bateman@kcl.ac.uk

**Data Availability Statement:** RNA-Seq data have been deposited in NCBI's Gene Expression Omnibus and are accessible through GEO Series

## Abstract

Mutations in subunits of the mitochondrial NADH dehydrogenase cause mitochondrial complex I deficiency, a group of severe neurological diseases that can result in death in infancy. The pathogenesis of complex I deficiency remain poorly understood, and as a result there are currently no available treatments. To better understand the underlying mechanisms, we modelled complex I deficiency in *Drosophila* using knockdown of the mitochondrial complex I subunit ND-75 (NDUFS1) specifically in neurons. Neuronal complex I deficiency causes locomotor defects, seizures and reduced lifespan. At the cellular level, complex I deficiency does not affect ATP levels but leads to mitochondrial morphology defects, reduced endoplasmic reticulum-mitochondria contacts and activation of the endoplasmic reticulum unfolded protein response (UPR) in neurons. Multi-omic analysis shows that complex I deficiency dramatically perturbs mitochondrial metabolism in the brain. We find that expression of the yeast non-proton translocating NADH dehydrogenase NDI1, which reinstates mitochondrial NADH oxidation but not ATP production, restores levels of several key metabolites in the brain in complex I deficiency. Remarkably, NDI1 expression also reinstates endoplasmic reticulum-mitochondria contacts, prevents UPR activation and rescues the behavioural and lifespan phenotypes caused by complex I deficiency. Together, these data show that metabolic disruption due to loss of neuronal NADH dehydrogenase activity cause UPR activation and drive pathogenesis in complex I deficiency.

## Author summary

Mutations in the mitochondrial NADH dehydrogenase cause complex I deficiency, a mitochondrial disease characterised by severe neurological problems and death in the first

accession number GSE214562 (https://www.ncbi.nlm.nih.gov/geo/query/acc.cgi?acc= GSE214562).

**Funding:** This work was funded by Alzheimer's Research UK (ARUK-IRG2017A-2) and the MRC (MR/V013130/1) to JMB; LG was supported by the UK Medical Research Council (MR/N013700/1) and King's College London MRC Doctoral Training Partnership in Biomedical Sciences; STS and LFA were supported by a Motor Neurone Disease Association (UK) PhD studentship (Sweeney/ Oct15/884-792) and a BBSRC project grant (BB/ M002322/1) to STS; R.P.C was supported by a Northwestern University Pulmonary and Critical Care Department Cugell predoctoral fellowship. The funders had no role in study design, data collection and analysis, decision to publish, or preparation of the manuscript.

**Competing interests:** The authors have declared that no competing interests exist.

years of life. To understand the underlying mechanisms, we modelled complex I deficiency in the fruit fly *Drosophila*. Flies with complex I deficiency in neurons have problems with movement, seizures and severely reduced lifespan. Complex I deficiency in *Drosophila* neurons causes altered mitochondrial morphology and reduced contacts between the mitochondria and endoplasmic reticulum but does not affect ATP levels. Moreover, a stress signalling pathway called the unfolded protein response (UPR) is activated in complex I deficient neurons. Complex I deficiency also alters metabolism in the brain. Remarkably, restoring the NADH dehydrogenase activity but not the proton pumping ability of complex I in neurons, by expressing the yeast NDI1 enzyme, restores mitochondrial morphology, prevents UPR activation and rescues the behavioural and lifespan phenotypes in complex I deficient flies. Our data suggest that metabolic disruption due to loss of neuronal NADH dehydrogenase activity drive pathogenesis in complex I deficiency.

## Introduction

Mitochondrial NADH dehydrogenase-ubiquinone oxidoreductase (complex I) plays essential roles in oxidative phosphorylation (OXPHOS), NADH oxidation and reactive oxygen species (ROS) generation. Mutations in complex I subunits cause complex I deficiency, a clinically heterogenous disorder that frequently results in severe neurological symptoms and death in infancy [1]. Mutations in the complex I genes NDUFS4 and NDUFS1 commonly lead to Leigh syndrome, an aggressive form of complex I deficiency [2]. In mice, knockout of *Ndufs4* specifically in the brain recapitulates the severe motor, seizure, neuroinflammatory and lifespan phenotypes associated with Leigh syndrome [3], indicating that symptoms are driven by loss of complex I activity in the nervous system.

Perturbed mitochondrial function triggers mitochondrial stress signalling pathways. Mitochondrial stress signalling enables mitochondria to communicate with the nucleus and reprogram nuclear gene expression [4–6]. Mitochondrial stress signalling can act through different mechanisms, depending on the context. In higher eukaryotes, a common target of mitochondrial stress signalling is activating transcription factor 4 (ATF4) [7–9]. ATF4 can be activated downstream of the endoplasmic reticulum unfolded protein response (UPR), or the integrated stress response (ISR). Importantly, targeting stress signalling pathways, including the UPR and ISR, has shown therapeutic potential in animal models [10,11].

Animal models of mitochondrial disease have provided valuable insights into disease mechanisms. Mouse and *Drosophila* complex I deficiency models have replicated several aspects of the human disease [3,12–16]. However, animal models have not previously been exploited to examine mitochondrial stress signalling in complex I deficiency. Here we interrogate a new neuron-specific *Drosophila* model of mitochondrial complex I deficiency. Knockdown of the complex I subunit ND-75 (NDUFS1) in neurons causes neurological phenotypes including climbing and locomotor defects, seizures and greatly reduced lifespan. ND-75 knockdown in neurons does not affect ATP levels but perturbs mitochondrial morphology, causes reduced endoplasmic reticulum (ER)-mitochondria contacts and triggers the UPR, including activation of ATF4. Loss of ND-75 also alters cellular metabolism and perturbs the TCA cycle. Expression of the single subunit yeast NADH dehydrogenase NDI1 in neurons, which facilitates the TCA cycle, rescues the aberrant mitochondrial morphology and loss of ER-mitochondria contacts resulting from ND-75 knockdown. UPR activation and ATF4 expression are also prevented by NDI1 expression. Moreover, NDI1 almost completely rescues the severe behavioural and

lifespan phenotypes caused by ND-75 knockdown. Metabolomic analysis shows that NDI1 expression reverses the changes in specific metabolites caused by ND-75 knockdown, including increased levels of the neurotransmitter GABA. Collectively, this study provides novel insight into the mechanisms underlying the neurological manifestations of mitochondrial complex I deficiency.

## Results

### Knockdown of ND-75 in Drosophila neurons models mitochondrial complex I deficiency

Mutations in *NDUFS1* cause complex I deficiency and Leigh syndrome [2,17]. Complex I initiates the electron transport chain, accepting electrons from NADH which then pass through the complex to ubiquinone. The subunits that form complex I are grouped into 3 modules: N, P and Q. NDUFS1 is an [Fe-S] cluster containing subunit involved in electron transfer that lies within the N module of the complex [18]. To model complex I deficiency in *Drosophila* we used a previously validated short hairpin RNAi (HMS00853) targeting the *Drosophila* NDUFS1 homolog ND-75 [19]. We confirmed that ND-75 knockdown (hereafter referred to as ND-75$^{KD}$) ubiquitously in adult flies using the GeneSwitch system [20] strongly reduced *ND-75* mRNA expression and caused an ~87% reduction in mitochondrial NADH dehydrogenase activity (S1A and S1B Fig). The ND-75 RNAi alone caused a significant reduction in ND-75 expression indicating leaking expression of the transgene (S1A Fig). Ubiquitous ND-75$^{KD}$ also significantly reduced the expression of ND-30 (S1C–S1E Fig), the orthologue of mammalian NDUFS3 and a component of Q module of complex I [21], suggesting collapse of the whole complex, and mirroring what has previously been observed with ND-75$^{KD}$ in *Drosophila* muscle [21].

Motor impairments are frequently reported in complex I deficiency patients [22]. Pan-neuronal ND-75$^{KD}$ using *nSyb-GAL4* caused late pupal lethality, but dampening GAL4 activity by combining *nSyb-GAL4* with *Tub-Gal80$^{ts}$* at 25°C resulted in viable adult flies that were almost completely unable to climb (Fig 1A). Similar results were obtained with an independent non-overlapping ND-75 short hairpin RNAi (HMS00854) (S1H Fig). Although the ND-75 RNAi was leaky and caused reduced ND-75 expression this had no effect on climbing ability, whereas ubiquitous ND-75$^{KD}$ caused a strong reduction in climbing ability (S1A, S1F, and S1G Fig).

To further analyse this locomotor phenotype we next measured open-field behaviour in pan-neuronal ND-75$^{KD}$ flies. The average speed and total distance moved by ND-75$^{KD}$ flies was dramatically reduced, and immobility was increased compared to control flies (Fig 1B–1E). Together, these data demonstrate that ND-75$^{KD}$ flies have severe motor dysfunction.

Alongside motor impairments, seizures are a common symptom of complex I deficiency [23,24]. Using a mechanical stress-induced seizure assay we found that pan-neuronal ND-75$^{KD}$ flies had a dramatic seizure phenotype, with 94% of flies developing seizures, which were significantly longer than controls and, in some cases, lasting four to five minutes (Fig 1F and 1G). Feeding problems are also common in complex I defiency patients [17] and we found that food intake was significantly reduced in ND-75$^{KD}$ flies (Fig 1H). Together with severe neurological symptoms, complex I deficiency patients typically die within the first few years of life [17,25]. Consistent with this, pan-neuronal ND-75$^{KD}$ flies had dramatically reduced lifespan, with a median survival of five days, compared to 48 days for controls (Fig 1I). Overall, these data show that neuronal-specific loss of ND-75 in *Drosophila* models the severe neurological manifestations of complex I deficiency.

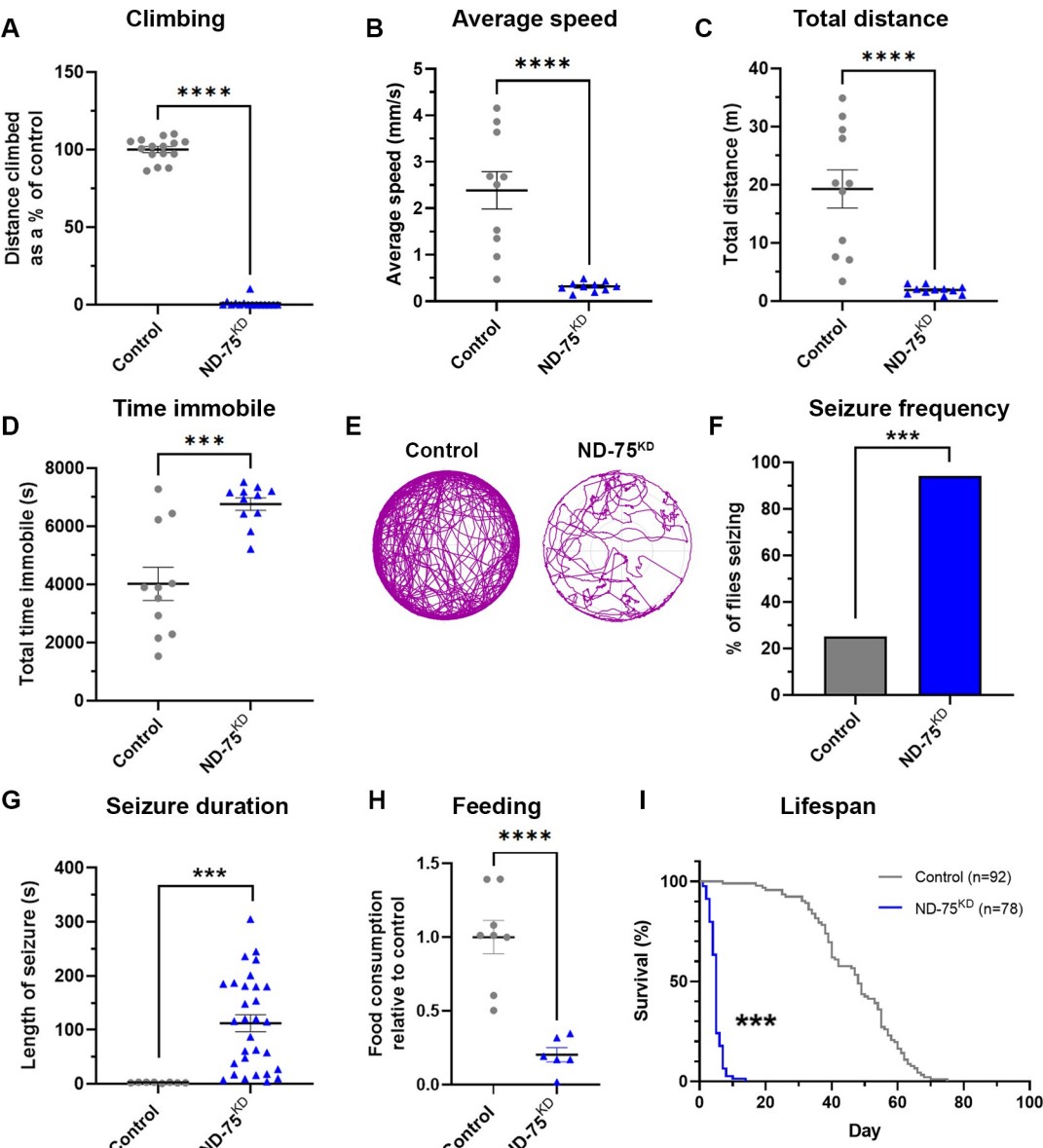

**Fig 1. ND-75**<sup>KD</sup> **flies have reduced mobility, seizures and decreased lifespan.** (A) ND-75<sup>KD</sup> flies are almost completely unable to climb. Control n = 15, ND-75<sup>KD</sup> n = 15 flies. (B-D) In an 135 min open field assay, ND-75<sup>KD</sup> flies have reduced average speed (B), move a shorter total distance (C) and spend more time immobile (D) Control n = 11, ND-75<sup>KD</sup> n = 11 flies. (E) Example open field track plot of control and ND-75<sup>KD</sup> flies. (F, G) ND-75<sup>KD</sup> flies have more seizures than controls (F), which last up 4–5 mins in some cases (G) Control n = 8, ND-75<sup>KD</sup> n = 30 flies. (H) Food consumption over 24 hours is reduced in ND-75<sup>KD</sup> flies. Control n = 8, ND-75<sup>KD</sup> n = 6 feeding assay containers. (I) Lifespan is greatly reduced in ND-75<sup>KD</sup> flies. ND-75 RNAi expressed using *Tub-Gal80*<sup>ts</sup>; *nSyb-Gal4*. Controls are *Tub-Gal80*<sup>ts</sup>; *nSyb-Gal4* hemizygotes. Control n = 92, ND-75<sup>KD</sup> n = 78. 1 day old male flies were used in (A)-(E), 1 day old males and females were used in (F)-(H). Males flies were used in (I). Data are represented as mean ± SEM and were analysed using the Student's unpaired t test, Chi-squared for seizure frequency, or log-rank test for survival curve. ***p < 0.001. ****p < 0.0001.

## ND-75<sup>KD</sup> causes reduced neuronal ER-mitochondrial contacts

Mitochondria in fibroblasts isolated from patients with complex I deficiency are fragmented and have significantly altered morphology [26,27]. We therefore investigated the effect of complex I deficiency on mitochondrial morphology in neurons. Using super-resolution

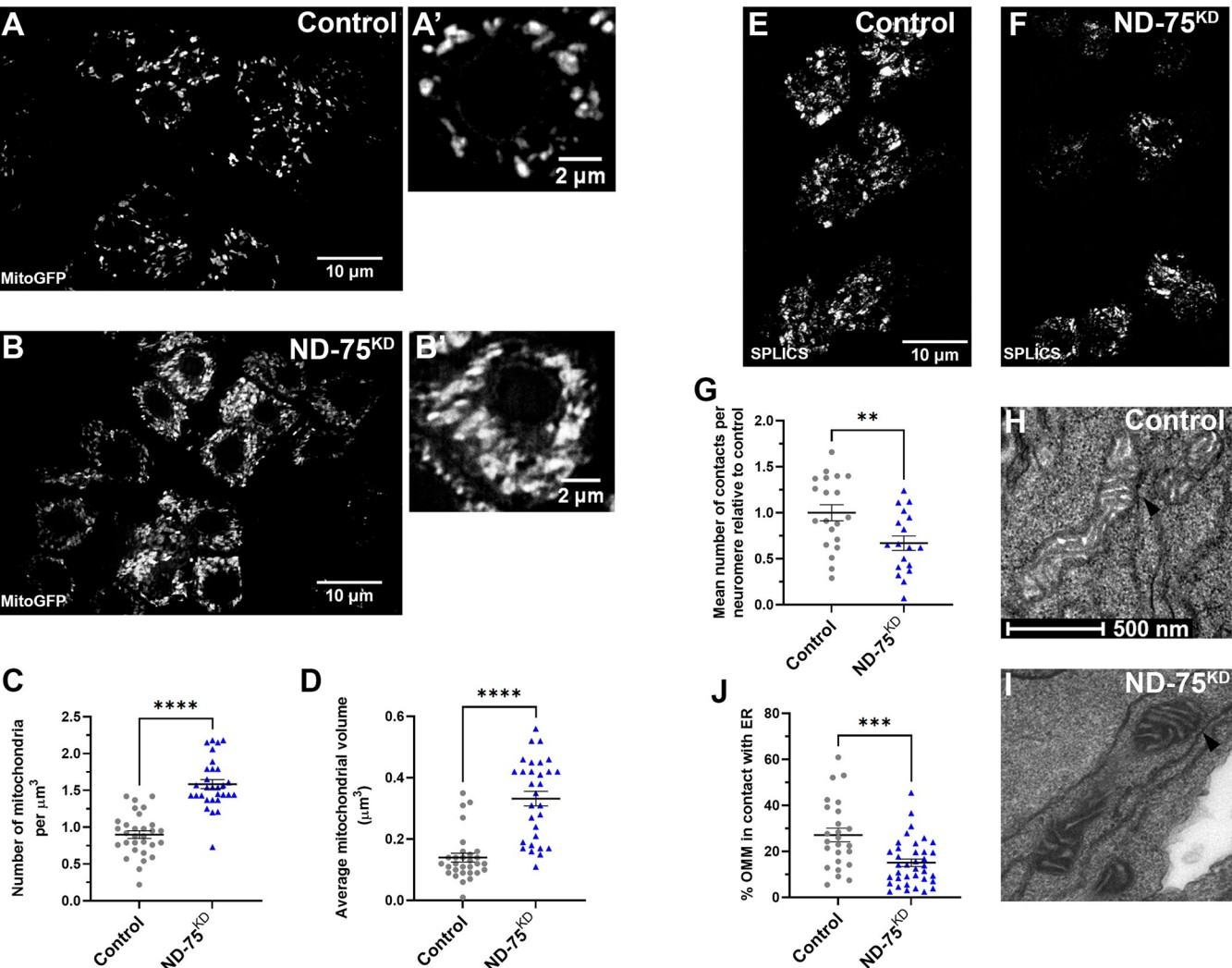

**Fig 2. ND-75^KD in larval neurons causes aberrant mitochondrial morphology and reduced ER-mitochondria contacts.** (A, B) Expression of mitochondria-targeted GFP (mitoGFP) to visualise mitochondria in control (A) or ND-75 knockdown (B) larval motor neurons using *OK371-Gal4*. Images were taken using iSIM. (C, D) Quantification of mitochondrial number and volume in larval motor neurons with ND-75 knockdown using *OK371-Gal4*. Control n = 30, ND-75^KD n = 30 ROIs (E, F) Visualisation of ER-mitochondria contacts by split-GFP-based contact site sensor (SPLICS) expression in control (E) or ND-75 knockdown (F) larval motor neurons using *OK371-Gal4*. (G) Quantification of SPLICS puncta. Control n = 20, ND-75^KD n = 18 neuromeres (H, I) Transmission electron microscopy images of mitochondria in larval CNS tissue from control or with pan-neuronal ND-75 knockdown (using *nSyb-Gal4*). Arrowheads indicate ER-mitochondria contacts. (J) Quantification of ER-mitochondria contacts. Control n = 25, ND-75^KD n = 37 mitochondria. Controls are *OK371-Gal4* or *nSyb-Gal4* hemizygotes. Data are represented as mean ± SEM and were analysed using the Student's unpaired t test. **p<0.01, ***p < 0.001, ****p < 0.0001.

microscopy, we found that ND-75^KD in larval motor neurons led to striking alterations to the normal reticular mitochondrial morphology a 76% increase in mitochondrial number and a significant increase in mitochondrial volume (Fig 2A–2D). At the ultrastructural level, neuronal ND-75^KD did not obviously perturb mitochondrial cristae morphology in larval neurons or the adult brain (Figs 2H, 2I, 6G, and 6I).

To determine the effect of ND-75 loss on ATP synthesis, we used the Förster resonance energy transfer (FRET)–based ATP biosensor AT-NL [28], which we previously validated [29]. In neurons, as expected, AT-NL produced a significantly higher FRET signal than the AT-RK control probe that does not sense ATP (S2A–S2C Fig). ND-75^KD did not alter the

AT-NL probe FRET signal in neurons (S2D–S2F Fig), indicating that loss of ND-75 does not affect ATP levels in neurons. Similarly, whole heads from flies with pan-neuronal ND-75[KD] did not show any change in ATP levels (S2G Fig).

Changes to mitochondrial morphology lead to alterations in ER-mitochondria contacts [30,31]. Depending on the cell-type, it is estimated that ~5–35% of the mitochondrial network surface is in close proximity to the ER [32,33]. At sites of contact, mitochondria and the ER physically interact through the formation of tethering complexes [34]. ER-mitochondria contacts play vital roles in processes including $Ca^{2+}$ homeostasis, mitochondrial fission, autophagy and lipid transfer. In neurons, disturbance of these contacts is strongly associated with altered cellular signalling and disease [35,36]. Therefore, we next assessed whether ER-mitochondria contacts are affected in ND-75[KD] neurons. To do this, we generated flies expressing the split-GFP-based contact site sensor (SPLICS) [31]. SPLICS consists of an ER membrane-bound moiety and a mitochondrial membrane-bound moiety, which reconstitute to form a functional GFP protein when the ER and mitochondria are close enough (8–50 nm) to establish contacts. Mitochondrial morphological changes that alter ER-mitochondrial contacts are detected by SPLICS probes [30,31]. In neuronal cell bodies our SPLICS probe localised to puncta that were significantly reduced by knockdown of the ER-mitochondria tethering protein IP$_3$R [37], validating this tool (S3A–S3C Fig). ND-75[KD] in larval motor neurons caused a significant reduction in the number of SPLICS puncta (Fig 2E–2G), suggesting reduced ER-mitochondria connectivity. We confirmed this finding in the larval nervous system at the ultrastructural level using transmission electron microscopy (Figs 2H–2J,). These data suggest that the dramatic alteration in mitochondrial morphology resulting from ND-75 knockdown significantly reduces ER-mitochondria contacts.

## The UPR is activated in ND-75[KD] flies

Disruption of ER-mitochondria contacts has detrimental consequences for ER function, triggering ER stress and activation of the unfolded protein response (UPR), a stress signalling pathway [38,39]. Furthermore, the UPR contributes to disease progression in animal models of neurodegenerative disease [11]. To investigate whether the UPR is activated in ND-75[KD] flies, we analysed expression of the UPR transcription factor ATF4. ATF4 is not normally expressed in the *Drosophila* brain but ND-75[KD] in neurons caused strong activation of ATF4 expression in the larval CNS and in the adult brain (Fig 3A, 3B, 3E and 3F). Induction of the UPR activates protein kinase R-like endoplasmic reticulum kinase (PERK), resulting in ATF4 up-regulation through the phosphorylation of the translation initiation factor eIF2α [39]. Consistent with this, ND-75[KD] increased phosphorylated eIF2α levels in neurons, and knockdown of PERK abrogated ATF4 activation by ND-75[KD] (Fig 3C, 3D and 3G–2J). Therefore, complex I deficiency activates ATF4 via the UPR in neurons.

To independently test the consequences of UPR activation in the nervous system we expressed dominant negative forms of the ER chaperone Hsc70-3 (GRP78, BiP), which ectopically activates the UPR [40]. Pan-neuronal expression of Hsc70-3[D231S] caused pupal lethality, while expression of Hsc70-3[K97S] caused a climbing defect (S3D Fig). This finding is consistent with the idea that activation of the UPR contributes to neuronal dysfunction in ND-75[KD] flies.

## Transcriptional and metabolic disruption in ND-75[KD] flies

Mitochondrial stress signalling enables mitochondria to reprogramme nuclear gene expression, impacting a wide range of cellular functions [5]. To gain insight into the transcriptional changes induced by mitochondrial complex I deficiency in neurons we performed transcriptomic analysis of head tissue from ND-75[KD] flies. The expression of 1694 genes were

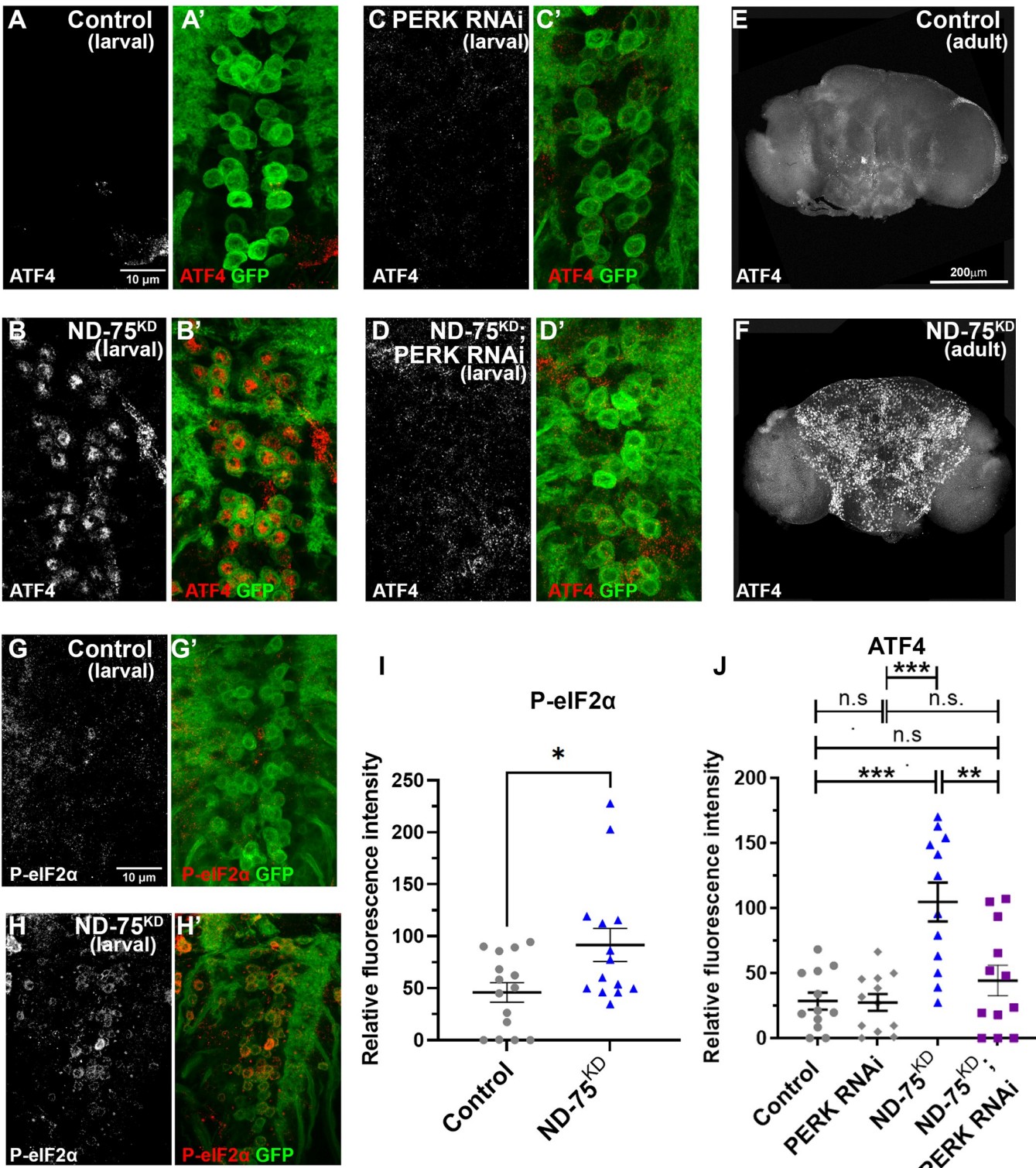

**Fig 3. ND-75^KD in larval and adult neurons activates the UPR.** (A-D) ATF4 (red) expression in control (A) or ND-75^KD (B), PERK knockdown (C), or ND-75^KD with PERK knockdown (D) larval motor neurons using *OK371-Gal4*. CD8-GFP (green) expression labels motor neurons. (E, F) ATF4 expression in control (E) or pan-neuronal ND-75^KD (F) adult brain tissue using *Tub-Gal80^ts*; *nSyb-Gal4*. (G, H) Phospho-eIF2α (P- eIF2α, red) expression in control (G) or ND-75^KD (H) larval motor neurons using *OK371-Gal4*. CD8-GFP (green) expression labels motor neurons. (I) Quantification of Phospho-eIF2α expression. Control n = 15, ND-75^KD n = 14 larval CNS. Student's t test. (J) Quantification of ATF4 expression. n = 12 larval CNS for all genotypes. 1 day old male and female flies were used in all panels. Data were analysed using ANOVA with Tukey's posthoc test. Controls are *OK371-Gal4* or *Tub-Gal80^ts*; *nSyb-Gal4* hemizygotes. Data are represented as mean ± SEM. n.s. not significant, *p<0.05, **p<0.01, ***p < 0.001.

significantly increased, and 2017 genes were significantly decreased in ND-75[KD] flies (S1 and S2 Tables), indicating a strong mitochondrial stress induced transcriptional response. Gene ontology (GO) analysis of genes with increased expression in ND-75[KD] flies showed significant enrichment of cellular components and molecular functions including transcriptional activity and ubiquitination (S3E and S3F Fig and S3 Table). However, apart from one of the two *Drosophila* orthologs of *Hsp10* (*CG9920*) whose expression was increased (S1 Table), the expression of genes that respond to the mitochondrial unfolded response (UPR[mt]; *Hsp60C*, *Hsc70-5* and *Hsp10*; [41]) were unchanged in ND-75[KD] flies, suggesting that the UPR[mt] is not activated. By contrast, consistent with activation of the UPR and ATF4 in ND-75[KD] neurons, 39 of the 230 high confidence *Drosophila* homologs of mammalian ATF4 target genes had significantly increased expression in ND-75[KD] flies, including the established ATF4 target *protein phosphatase 1 regulatory subunit 15* (*PPP1R15*), or Gadd34, which mediates dephosphorylation of eIF2α (S1 and S5 Tables) [42]. The expression of several autophagy genes (including Atg1, Atg2, Atg5, Atg8a) were also increased in ND-75[KD] flies, suggesting induction of autophagy and/or mitophagy (S1 Table).

GO analysis of genes with reduced expression showed striking enrichment for mitochondrial genes (Fig 4A). Moreover, Kyoto encyclopaedia of genes and genomes (KEGG; [43]) pathway analysis of genes with reduced expression showed genes involved in oxidative phosphorylation and the TCA cycle as significantly enriched (Fig 4B). Strikingly, in addition to ND-75, the expression of 28 genes encoding core or accessory subunits of complex I, or complex I assembly factors were significantly decreased in ND-75[KD] fly head tissue (Fig 4C and S2 Table), while none were significantly increased (S1 Table). Furthermore, expression of multiple subunits of the other four mitochondrial respiratory complexes were significantly decreased by neuronal ND-75[KD] (Fig 4C and S2 Table). Moreover, the expression of genes encoding numerous antioxidant enzymes were significantly decreased in ND-75[KD] brain tissue (S2 Table). These data reveal a concerted transcriptional response to depress neuronal mitochondrial respiratory complex gene expression in the brain in response to complex I deficiency.

Genes encoding TCA cycle enzymes isocitrate dehydrogenase, succinyl-CoA synthetase, succinate dehydrogenase, fumarate hydratase, malate dehydrogenase, citrate synthase and components of the pyruvate dehydrogenase complex and oxoglutarate dehydrogenase complex were significantly decreased in ND-75[KD] head tissue (Fig 4C and S2 Table), whilst no TCA cycle genes were significantly increased (S1 Table), strongly implying that neuronal mitochondrial metabolism is perturbed by neuronal complex I deficiency. To investigate this further, we performed metabolomic analysis of head tissue to directly analyse the mitochondrial metabolic changes in the brain caused by complex I deficiency (S6 Table). Principle component analysis showed that the metabolite profile in ND-75[KD] fly heads was markedly different to controls (Fig 4D). Moreover, ND-75[KD] flies had a significant increase in the level of 22 metabolites, and a significant decrease in the level of 21 metabolites compared to controls (S7 Table). Metabolite set enrichment analysis (MSEA) showed significant dysregulation of a variety of KEGG pathways in ND-75[KD] head tissue including purine and amino acid metabolism, the TCA cycle and urea cycle (Fig 4E). Levels of the purine nucleotide adenine were significantly increased, and the TCA cycle intermediate fumarate were significantly decreased in ND-75[KD] heads (Fig 4F–4H and S7 Table). 2-hydroxyglutarate levels were strongly increased (Fig 4F and S7 Table), which we and others have shown is elevated in other mitochondrial dysfunction/disease models [9,44]. Glutamate and γ-aminobutyric acid (GABA) levels were also increased in ND-75[KD] fly heads (Fig 4I and S7 Table), indicating that neuronal complex I deficiency affects neurotransmitter levels. Collectively, these data demonstrate that complex I deficiency in neurons disrupts metabolism.

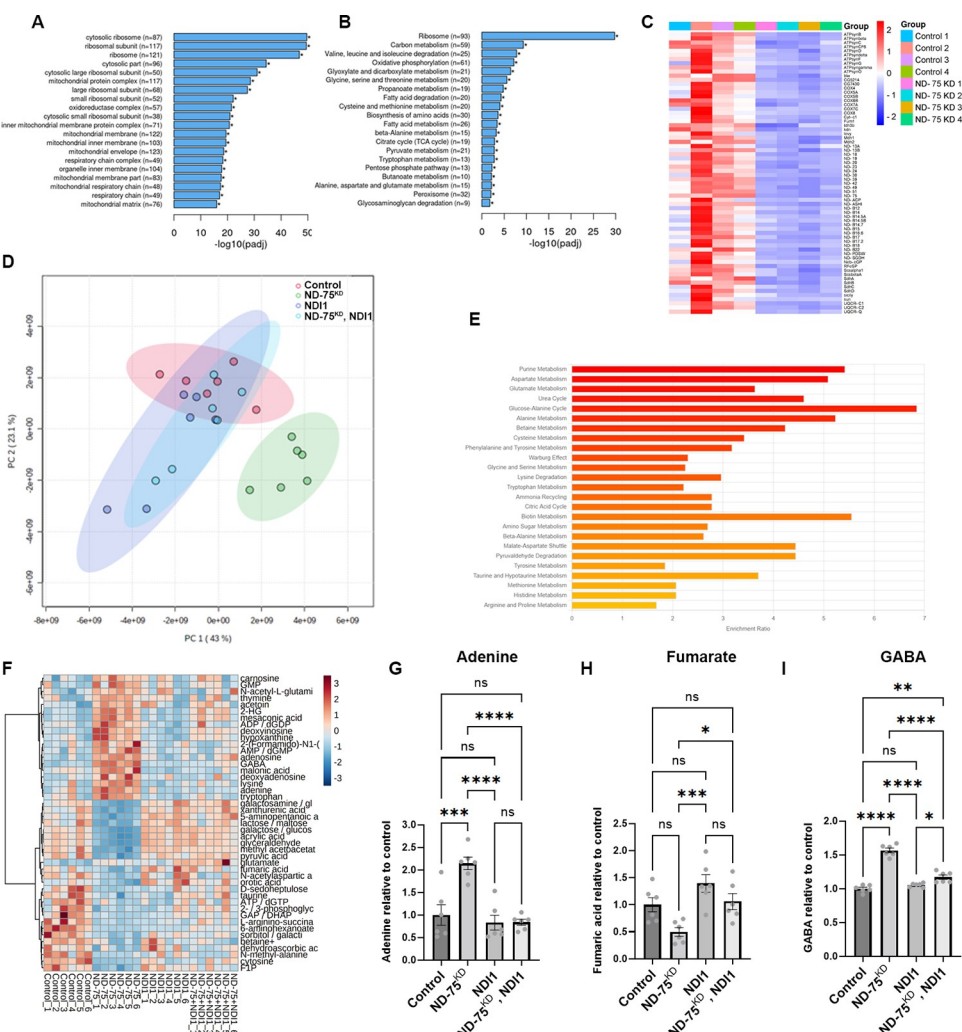

**Fig 4. ND-75^KD inhibits mitochondrial gene expression and disrupts the metabolism the brain.** (A) GO cellular component analysis of genes with significantly decreased expression in head tissue from pan-neuronal ND-75^KD using *Tub-Gal80^ts*; *nSyb*-Gal4. (B) KEGG pathway analysis of genes with significantly decreased expression in head tissue from pan-neuronal ND-75^KD using *Tub-Gal80^ts*; *nSyb*-Gal4. (C) Heatmap showing reduced expression of OXPHOS and TCA cycle genes in ND-75^KD head tissue. (D) Principle component analysis of metabolite levels in head tissue from control, ND-75^KD, NDI1 and ND-75^KD, NDI1 expressing flies using *Tub-Gal80^ts*; *nSyb*-Gal4. (E) MSEA analysis of significantly altered metabolites in head tissue from pan-neuronal ND-75^KD using *Tub-Gal80^ts*; *nSyb*-Gal4. (F) Heat map showing that a subset metabolites whose are significantly different in ND-75^KD heat tissue are reversed by NDI1 expression. (G-I) Levels of adenine (G), fumarate (H) and GABA (I) control, ND-75^KD, NDI1 and ND-75^KD, NDI1 expressing flies from metabolomic analysis. n = 6 biological replicates for all genotypes. Controls are *Tub-Gal80^ts*; *nSyb-Gal4* hemizygotes. 2 day old male and female flies were used in all panels. Data are represented as mean ± SEM and were analysed using the Student's t test. ns not significant, *p<0.05, **p<0.01, ***p < 0.001, ***p < 0.0001.

## NDI1 expression partially reverses the neuronal metabolic disruption in ND-75^KD flies

NADH oxidation by complex I replenishes NAD+, an essential coenzyme for the TCA cycle enzymes isocitrate dehydrogenase, α-ketoglutarate dehydrogenase and malate dehydrogenase. TCA cycle intermediates contribute to numerous biosynthetic pathways, including amino acid, fatty acid and purine/pyrimidine synthesis [45,46]. Disruption of the TCA cycle could thus lead to wide-ranging alterations in metabolism in ND-75^KD flies. We therefore

hypothesised that impaired complex I NADH dehydrogenase activity is a key driver of neuronal dysfunction in ND-75[KD] flies. To test this hypothesis, we utilised the *Saccharomyces cerevisiae* single-subunit non-proton translocating NADH dehydrogenase NDI1, which oxidises mitochondrial NADH and transfers electrons to ubiquinone but does not directly contribute to OXPHOS-mediated ATP production [47]. NDI1 expression should therefore restore TCA cycle activity in ND-75[KD] neurons deficient for complex I activity. We first confirmed that addition of the NDI1 transgene did not alter the strong reduction in ND-75 expression in ND-75[KD] flies with NDI1 expression (S1A Fig). Metabolomic analysis of heads from flies with pan-neuronal ND-75[KD] combined with NDI1 expression showed a dramatically altered metabolite profile compared to ND-75[KD] alone (Fig 4D). Expression of NDI1 reversed the 23 of 45 metabolites whose levels were aberrant in ND-75[KD] head tissue, including adenine and fumarate (Fig 4F–4I and S3 and S4 Tables). Interestingly, NDI1 expression also reversed the dramatic increase in GABA levels in ND-75[KD] flies (Fig 4F and 4I and S3 and S4 Tables). Thus, expression of NDI1 prevents many of the metabolic changes caused by knockdown of ND-75 in neurons, suggesting that restoration of mitochondrial NADH dehydrogenase activity broadly benefits neuronal metabolism in complex I deficiency.

### NDI1 expression reinstates ER-mitochondria contacts, rescues neuronal dysfunction and death, and prevents UPR activation caused by complex I deficiency

As expected, expression of NDI1 restored the loss of NADH dehydrogenase activity in ND-75[KD] flies (Fig 5A). To test whether impaired NADH dehydrogenase-associated metabolic defects underlie the neuronal dysfunction caused by complex I deficiency, we next analysed the effects of NDI1 expression on motor function, seizure susceptibility and lifespan. NDI1 expression dramatically improved the climbing defect and completely prevented seizures in ND-75[KD] flies (Fig 5B and 5D). Moreover, ND-75[KD] flies expressing NDI1 had a median lifespan of 51 days, similar to controls (54 days; Fig 5E).

Complex I is a major source of ROS and NDI1 has been shown to increase reverse electron transport-mediated ROS production at complex I through the enhancement of reverse electron transport [48]. We analysed ROS using a ratiometric fluorescent probe (mito-roGFP2-Grx1) to measure glutathione redox potential in neurons [49], which we have previously validated [29,50]. ND-75[KD] in neurons caused a decrease in probe oxidation (S4A, S4B, and S4E Fig), indicating reduced mitochondrial glutathione associated ROS. As expected, expression of NDI1 in neurons alone or in combination with ND-75 knockdown increased mitochondrial glutathione redox potential (S4A-S4E Fig). However, manipulation of ROS levels in ND-75[KD] flies through knock-down of Sod1, or Sod2, or overexpression of Sod2, catalase, or a mitochondrially targeted catalase, using previously validated transgenes [51], did not modify the climbing phenotype (S4F–S4J Fig). Therefore, although NDI expression elevates ROS levels, the severe behavioural phenotypes in ND-75 [KD] flies are unlikely to be the result of impaired ROS production.

Finally, following the observation that NDI1 expression rescues neuronal function in ND-75[KD] neurons, we investigated whether NDI1 expression ameliorates ND-75[KD] induced changes to mitochondrial morphology, ER-mitochondria contacts and activation of the UPR. Strikingly, we found that NDI1 expression rescued the mitochondrial number and morphology changes (Fig 6A–6F), the loss of ER-mitochondria contacts (Fig 6G–6K) and prevented ATF4 activation caused by ND-75 knockdown in neurons (Fig 6L–6P). Therefore, using NDI1 to at least partially reverse the metabolic disruption caused by ND-75 knockdown prevents UPR activation, neuronal dysfunction and early death in *Drosophila*.

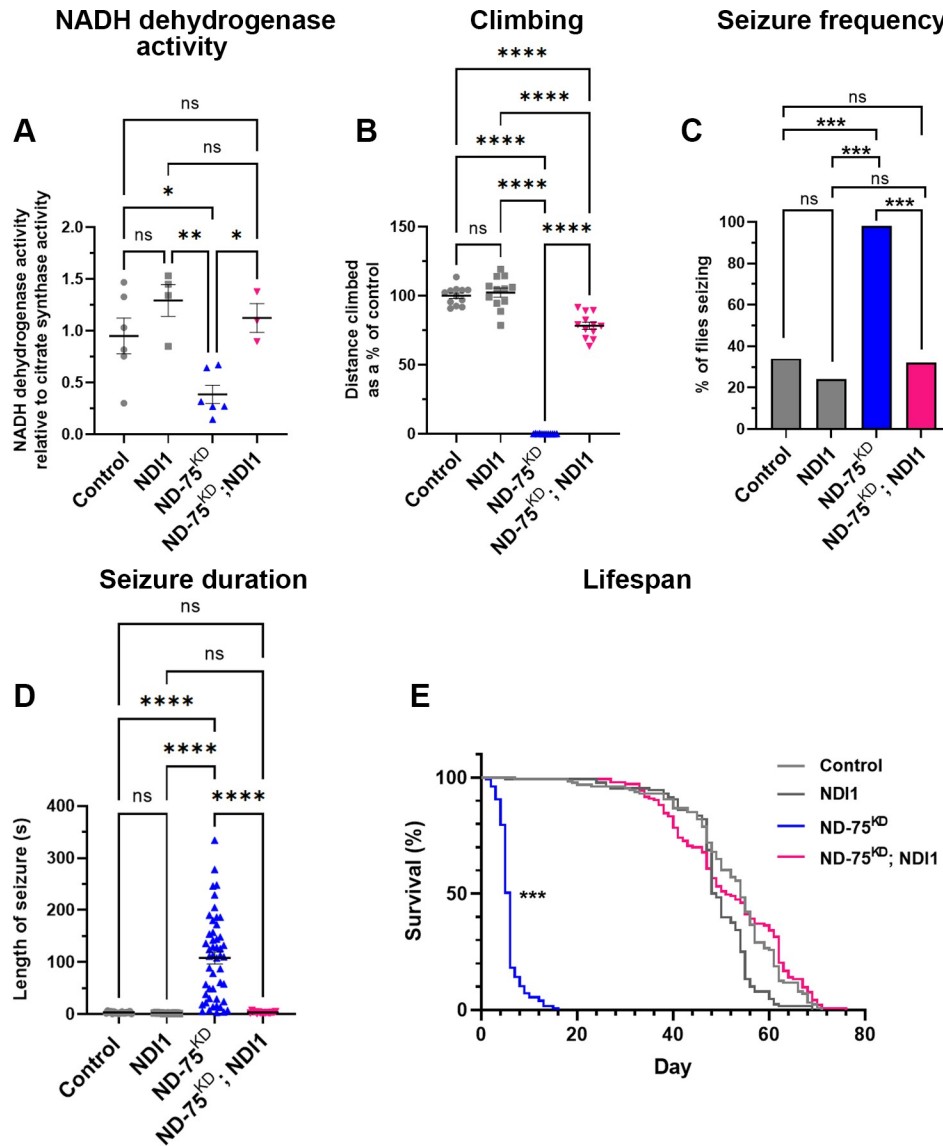

**Fig 5. Neuronal NDI1 expression restores climbing abllity, prevents seizures and early death in ND-75^KD flies.**
(A) Complex I activity in mitochondria isolated from adult flies with ubiqitious NDI1 expression, ND-75^KD, or ND-75^KD with NDI1 using *Da-GS-Gal4*. n = 5 biological replicates for all genotypes. Control n = 6, NDI1 n = 4, n = ND-75^KD n = 6, ND-75^KD with NDI1 n = 3 biological replicates. (B) Quantification of climbing ability in control flies or with pan-neuronal expression of NDI1, ND-75^KD, or ND-75^KD with NDI1. n = 12 flies for all genotypes. (C, D) Quantification of seizure frequency (C) and duration (D) in control flies or with pan-neuronal expression of NDI1, ND-75^KD, or ND-75^KD with NDI1. Control n = 17, NDI1 n = 12, n = ND-75^KD n = 48, ND-75^KD with NDI1 n = 16 flies. (E) Lifespan analysis in control flies or with pan-neuronal expression of NDI1, ND-75^KD, or ND-75^KD with NDI1. ND-75 RNAi expressed using *Tub-Gal80^ts*; *nSyb-Gal4*. Controls are *Tub-Gal80^ts*; *nSyb-Gal4* hemizygotes. Control n = 130, NDI1 n = 128, ND-75^KD n = 127, ND-75^KD; NDI1 n = 152. 5 day old male and female flies were used in (A). 1 day old male flies were used in (B) and 1 day old males and females in (C) and (D). Male flies were used in (E). Data are represented as mean ± SEM and were analysed using ANOVA with Tukey's posthoc test, Chi-squared for seizure frequency, or log-rank test for survival curve. n.s not significant, ***p < 0.001.

## Discussion

Neurons are highly energetically and metabolically demanding and so are acutely sensitive to mitochondrial dysfunction. We have dissected the cellular and molecular mechanisms

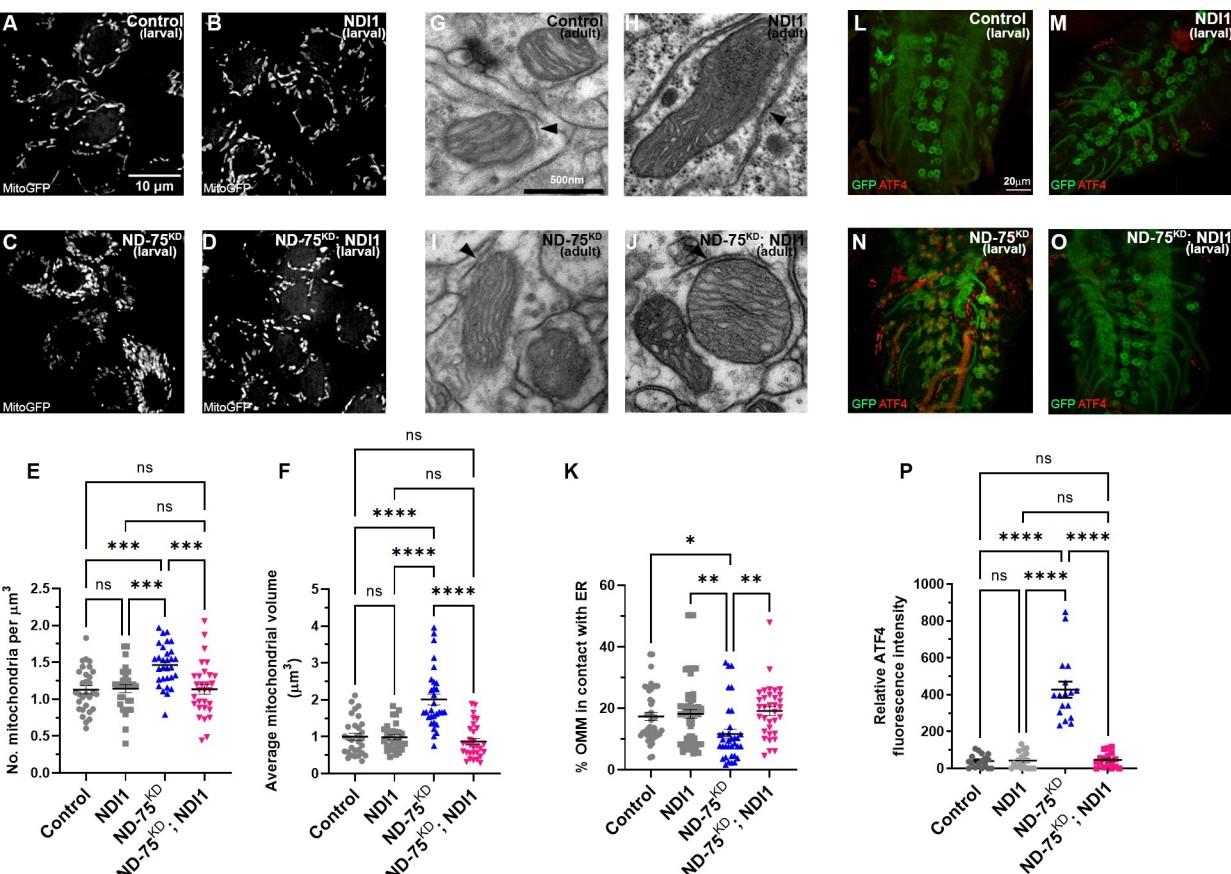

**Fig 6. NDI1 expression reverses the mitochondrial defects and prevents UPR activation in ND-75^KD neurons.** (A-D) Mitochondria-targeted GFP (MitoGFP) expression in control (A), NDI1 expression (B), ND-75^KD (C), or ND-75^KD with NDI1 (D) larval motor neurons using *OK371-Gal4*. Images taken using iSIM. (E) Quantification of mitochondrial number. n = 30 ROIs for all genotypes. (F) Quantification of mitochondrial volume. n = 30 ROIs for all genotypes. (G-J) Transmission electron microscopy images of mitochondria in adult brain tissue from control (G), or pan-neuronal NDI1 expression (H), ND-75^KD (I), or ND-75^KD with NDI1 (J) using *Tub-Gal80^ts*; *nSyb-Gal4*. Arrowheads indicate ER-mitochondria contacts. (K) Quantification of ER-mitochondria contacts. Control n = 44, NDI1 n = 48, n = ND-75^KD n = 35, ND-75^KD with NDI1 n = 34 mitochondria. 2 day old male and female flies were used. (L-O) ATF4 expression (red) in control (L), NDI1 expression (M), ND-75^KD (N), or ND-75^KD with NDI1 (O) larval motor neurons using *OK371-Gal4*. CD8-GFP expression (green) labels motor neurons. (P) Quantification of ATF4 expression. ANOVA with Tukey's posthoc test. n = 17 larval CNS for all genotypes. Controls are *OK371-Gal4* or *Tub-Gal80^ts*; *nSyb-Gal4* hemizygotes. Data are represented as mean ± SEM. n.s. not significant, *p<0.05, **p<0.01, ***p < 0.001, ***p < 0.0001.

underlying neuronal complex I deficiency *in vivo* using ND-75^KD in *Drosophila*. Loss of complex I activity in neurons has profound effects on mitochondrial morphology and decreases ER-mitochondria contacts. Consistent with disruption of ER-mitochondria contacts, we find activation of the ER UPR and expression of ATF4 in ND-75^KD neurons. ND-75^KD causes transcriptional reprogramming, potentially as an attempt to mitigate the loss of complex I activity, repressing complex I and TCA cycle gene expression. There is also evidence of alterations in numerous metabolic pathways, beyond the mitochondrion, as a result of complex I deficiency. Expression of yeast NDI1 in ND-75^KD flies reverses key metabolic changes, suggesting that restoring mitochondrial metabolism prevents UPR activation and neuronal dysfunction caused by complex I deficiency.

A previous *Drosophila* ND-75 knockdown model used a different ND-75 RNAi line, which results in much weaker phenotypes [13]. Using the same ubiquitous Gal4 driver (Da-Gal4) Hegde et al., found ND-75 knockdown resulted in adult flies with reduced lifespan, whereas

we found that ubiquitous ND-75 knockdown caused developmental lethality. Moreover, Hegde et al., found neuronal specific ND-75 knockdown flies had a 25 day median lifespan, whereas our ND-75[KD] model has a median lifespan of 5 days. Although we have not directly compared the efficiency of the two RNAi lines, the differences are likely due to the highly efficient knockdown of ND-75 expression in our ND-75[KD] model.

ATP levels are unaffected by ND-75 knockdown either in individual neurons, or in ND-75[KD] fly whole head tissue. These findings contrast with a previous *Drosophila* mitochondrial disease model with a mutation in the mitochondrial-DNA-encoded *NADH dehydrogenase subunit 2* (*ND2*) gene [52]. *ND2* mutants have reduced ATP levels in heads from 35 day old flies, whereas ATP levels in young flies were not analysed. We analysed ATP levels in very young (2 day old) ND-75[KD] flies, at which time the brain may have more metabolic plasticity than in old flies. Consistent with this, metabolomic data show significantly increased lactate levels in ND-75[KD] head tissue (S2H Fig and S6 Table).

Understanding the basis of the neurological manifestations of complex I deficiency in patients is hampered by the complex interactions between neurons and glia and the systemic effects of changes to tissues and organs outside the nervous system. Our ND-75[KD] model provides cell autonomous insight into the effects of neuronal complex I deficiency *in vivo*. Neuronal ND-75[KD] causes mitochondrial morphology changes, similar to those caused by knockout of the mitochondrial transcription factor TFAM in dopaminergic neurons [53]. The dramatic reduction in expression of respiratory and TCA cycle genes caused by ND-75[KD] unbalances the expression of mitochondrial genes encoded by the nuclear and mitochondrial genomes. This disruption of inter-organelle gene expression may underlie the mitochondrial morphology defects in ND-75[KD] neurons. Interestingly, the master regulator of mitochondrial biogenesis PGC-1α is regulated by the NAD+-dependent deacetylase SIRT1 [54], providing a potential link between the loss of NADH dehydrogenase activity and mitochondrial biogenesis in ND-75[KD] neurons.

Modulation of mitochondrial morphology driven by Drp1 overexpression causes reduced ER-mitochondria contacts in mammalian cultured cells and primary neurons [30,31]. The reduced ER-mitochondria contacts in ND-75[KD] neurons are therefore also likely caused by the dramatic mitochondrial morphological changes. ER-mitochondria tethering proteins are involved in regulation of the ER UPR, and modification of ER-mitochondria tethers can trigger UPR activation [55,56]. Together with these studies, our data suggest that ER UPR activation is caused by reduced ER-mitochondrial contacts in ND-75[KD] neurons. Altered ER-mitochondria contacts are a common feature of neurodegenerative disease and there is strong evidence to suggest that they play a pivotal role in disease pathogenesis [56]. Restoring ER-mitochondria contacts may therefore also have therapeutic potential in complex I deficiency.

Our data show that ND-75[KD] activates ATF4 expression via the ER UPR in neurons. ER UPR activation likely contributes to the neurological defects in ND-75[KD] flies as ectopic ER UPR activation via expression of dominant negative forms of Hsc70-3 also causes neuronal dysfunction. Activation of the ER UPR and integrated stress response (ISR) are frequently observed in response to mitochondrial dysfunction [4,5,7]. We have previously shown ER UPR and ATF4 activation due to mitochondrial dysfunction in *Drosophila* neurons using a TFAM overexpression model [9]. Loss of Pink1, Parkin, DJ-1 or OXPHOS inhibition also cause ER UPR or ISR activation [8,36,57,58]. The direct cause of ER UPR and ISR activation in cells with dysfunctional mitochondria is much less well understood. Recently, screening approaches in cultured mammalian cells identified a novel OMA1-DELE1-HRI pathway that activates the ISR in response to mitochondrial dysfunction [59,60]. Whether this pathway is active in neurons is not known but *Drosophila* do not have an HRI kinase homolog, so this mechanism is unlikely to be relevant to stress signalling in *Drosophila*.

NDI1 expression has been shown to extend lifespan and rescue the lethality caused by ubiquitous complex I deficiency in *Drosophila* [61,62]. Viral expression of NDI1 into the *substantia nigra* protects against rotenone induced toxicity in a rat model of Parkinson's disease [63]. In ageing models, NDI1 expression extends lifespan by increasing ROS through over reduction of the ubiquinone pool [48]. In the *Ndufs4* knockout mouse model of Leigh syndrome, NDI1 expression rescues the lifespan but not motor defects, potentially through regenerating mitochondrial NAD+ from NADH [16]. ND-75 knockdown in neurons does not affect ATP levels but our metabolic analysis shows that complex I inhibition causes dramatic changes in metabolism, potentially due to aberrant NAD+/NADH inhibiting the TCA cycle leading to broad metabolic alterations. We identified a variety of metabolites whose altered levels in complex I deficiency are reversed by NDI1 expression, including the TCA cycle metabolite fumarate. Increased levels of the major inhibitory neurotransmitter GABA may also contribute to the behavioural phenotypes in ND-75[KD] flies. Interestingly, *glutamate decarboxylase 1* (*Gad1*) expression is significantly increased by ND-75[KD] (S1 Table), suggesting that GABA levels may increase in part due to transcriptional mis-regulation. Overall, our study indicates that disruption of metabolism due to loss of NADH dehydrogenase activity is a driver of pathogenesis in the ND-75[KD] *Drosophila* model. The in-depth knowledge of the mechanisms contributing to neuronal complex I deficiency afforded by the *Drosophila* ND-75[KD] model may in future lead to therapeutic strategies tailored to combat the neurological manifestations in mitochondrial disease.

## Materials and methods

### Fly strains and growth conditions

Fly food consisted of 20g cornmeal (Dunn's River), 100g Brewer's yeast (MP biomedicals) and 8g Agar (Acros Organics) in 1L total volume of Milli-Q water. Ingredients were mixed and heated for 1.5 hours using a Systex Mediaprep machine. Once the mixture had cooled to 50˚C, 25 ml 10% w/v Nipagin (Sigma) dissolved in 70% ethanol, and 5ml propionic acid (Acros Organics) were added. After an additional 2 minutes of mixing, the liquid food was poured into vials and left to solidify. To produce RU-486 containing food, 2ml of 100mM RU-486 was added and mixed into 1L of liquid fly food immediately before pouring. Flies were maintained at 25˚C in a 12 hour light/dark cycle unless stated otherwise.

Details of fly stocks are listed in S8 Table. ND-75 RNAi lines were from the TRiP collection [64]. Fly stocks were obtained from the Bloomington Stock Center, the Vienna *Drosophila* Resource Center [65], the NIG-Fly Stock Center, Japan. For ubiquitous ND-75 knockdown using Daughterless-GeneSwitch-GAL4 [20], progeny were collected on the day of eclosion and kept at 29˚C for 5 days, to maximise Gal4 activity, on food containing 200μM RU-486. Pan-neuronal ND-75 knockdown was performed using *Tub-Gal80^{ts}*; *nSyb-Gal4* at 25˚C as ND-75 knockdown with *nSyb-Gal4* causes late pupal lethality. For imaging experiments embryos were laid over a 24 hour period at 25˚C, incubated for a further 24 hours at 25˚C, then incubated at 29˚C for three days prior to analysis, unless stated otherwise.

### Generation of SPLICS transgenic flies

To generate SPLICS flies, the SPLICS_L insertion was sub-cloned from a pcDNA3 backbone (gift from Tito Calí; [31]) into a pUASP vector as a BamHI-Xbal fragment. The pUASP-SPLICS construct was then integrated into the *Drosophila* genome by P-element mediated transposition. Embryo injections were performed by the Fly Facility of the Department of Genetics of the University of Cambridge.

## Behavioural analysis

Climbing assays were performed using male flies as described previously [9].

For assays using flies with neuronal dysfunction vials were placed on their sides during eclosion to prevent flies from becoming stuck in the food. For open-field locomotor activity flies were anaesthetised using $CO_2$ on the day of eclosion, and males of the desired genotype collected. Locomotor assays were performed the following morning, 1–4 hours after the start of the 12h light cycle. Flies were briefly anaesthetised on ice and placed into individual open-field arenas 35mm in diameter and 1.8mm in height with vibrating motors attached [66]. Flies were left to acclimatise to the arenas for 15 minutes prior to the start of video recording. Flies were video recorded for a total of 2 hours 15 minutes: after an initial rest period of 30 minutes, flies were subjected to a mechanical stimulus (5 vibrations, 0.2 seconds long, 0.5 seconds apart) followed by 15 minutes recovery. This pattern of stimulus and recovery was repeated a further 5 times before a final rest period of 15 minutes. Tracking of individual flies and analysis of average speed, total distance and immobility was performed using Anymaze software (Stoelting).

Mechanical stress-induced seizure assays were performed using both male and female flies. Flies of the desired genotype were collected on the day of eclosion and added to fresh vials so that each vial contained a total of 2 flies. The following day, flies were transferred to an empty vial, and vortexed at full speed for 10 seconds. Immediately after vortexing, flies were observed for seizures. The 'bang-sensitive' seizure phenotype triggered by mechanical stress is well defined [67]. A seizure was defined as a period of paralysis, potentially interspersed with limb twitching, wing movements and abdominal contractions. The length of each seizure was recorded, with the seizure considered to have ended when the fly stood upright. The experimenter was blinded to phenotype until data analysis had been completed.

Lifespan assays were performed using male flies. Male flies of the desired genotype were collected on the day of eclosion and added to fresh vials. 10 flies were initially added to each vial, and flies were incubated on their sides at 25°C for the duration of the assay. Every 2–3 days, flies were flipped into fresh vials and the number of dead flies in each vial were recorded. Dead flies that were carried over into the fresh vials were deducted from the next death count. Any flies that escaped during flipping, or were alive but stuck in the food so could not be flipped, were censored and deducted from the total fly count. Vials were flipped until all residing flies were dead.

To measure food intake, the CApillary FEeder assay (CAFE) was performed using 10 male and female flies per container for 24 hours as in [68].

## qRT-PCR

qRT-PCR was performed as described previously [9]. The following primers were used:

ND-75 forward: 5'-ACATTAACTACACGGGCAAGC-3'
ND-75 reverse: 5'- CAATCTCGGAGGCGAAAC-3'
Rpl4 forward: 5'-TCCACCTTGAAGAAGGGCTA-3'
Rpl4 reverse: 5'-TTGCGGATCTCCTCAGACTT-3'

## Mitochondrial NADH dehydrogenase (complex I) assay

Flies were snap-frozen in liquid nitrogen and stored at -70°C until required. For each replicate, 30 females and 30 males were used. Flies were first ground together using a Potter-Elvehjem type homogeniser at 700rpm for 12 passes, in 600μl extraction buffer. The extraction buffer consisted of 250mM sucrose, 5mM Tris (pH 7.4), 2mM EGTA (pH 8.0) and 1% bovine serum albumin (BSA) in ddH2O. The resulting homogenates next underwent a series of

centrifugation steps at 4˚C. Firstly, samples were spun at 1000g for 5 minutes. Supernatants were then taken and again spun at 1000g for 5 minutes. Supernatants were taken again and spun at 3000g for 10 minutes. Supernatants were then discarded, and 5ml extraction buffer was added to the remaining pellets. After another round of centrifugation at 3000g for 10 minutes, the supernatants were again discarded, and 5ml extraction buffer containing sucrose, Tris and EGTA only (no BSA), was added to the pellets. Samples were then spun for a final time at 7000g for 10 minutes. Supernatants were removed and pellets (mitochondrial fraction) were resuspended in 500 µl of extraction buffer without BSA. The concentration of mitochondria in resuspended pellets was determined using a Pierce bicinchoninic acid (BCA) assay (Thermo Fisher Scientific) following the manufacturer's protocol.

Mitochondrial complex I and citrate synthase activity were assessed using a published protocol [69]. To disrupt mitochondrial membranes and maximise enzymatic readouts, isolated mitochondria underwent 3 freeze-thaw cycles prior to use.

For measurement of rotenone-sensitive complex I activity, 30µg/ml isolated mitochondria were added to a cuvette containing 100µl potassium phosphate buffer (0.5M, 0.5M potassium phosphate monobasic added to 0.5M potassium phosphate dibasic, pH 7.5), 60µl fatty acid-free BSA (50mg/ml), 30µl freshly prepared potassium cyanide (10mM) and 10µl freshly prepared NADH (10mM). A second, identical cuvette was prepared, with the addition of 10µl rotenone (1mM). Volumes were then adjusted to 994µl using ddH$_2$O. After mixing, baseline absorbance was read for 2 minutes at 340nm and 25˚C using a SPECTROstar Nano (BMG Labtech). After recording of the baseline, 6µl ubiquinone1 (10 mM) was added to cuvettes, and the change in absorbance (Δabsorbance) was immediately recorded for 2 minutes at 340nm and 25˚C. To calculate the ubiquinone-dependent Δabsorbance, the baseline Δabsorbance was subtracted from Δabsorbance recorded after ubiquinone1 addition. The rotenone-sensitive Δabsorbance for each sample was then calculated by subtracting the Δabsorbance/minute with rotenone from the Δabsorbance/minute without rotenone. Rotenone-sensitive complex I activity was calculated using the following equation: enzyme activity (nmol/min/mg) = (Δabsorbance/min × 1,000)/[(extinction coefficient × volume of sample used in ml) × (sample protein concentration in mg/ml)], where the extinction co-efficient = 6.2 (NADH), volume = 1 and sample protein concentration = 0.03.

For measurement of citrate synthase activity, 5µg/ml isolated mitochondria were added to a cuvette containing 500µl Tris (200mM, pH 8.0) with 0.2% Triton X-100, 100µl freshly prepared 5,5-dithio-bis-(2-nitrobenzoic acid) (DNTB, 1mM) and 30µl acetyl CoA (10 mM). Volumes were then adjusted to 950µl using ddH$_2$O. After mixing, baseline absorbance was read for 2 minutes at 412nm at 25˚C using a SPECTROstar Nano (BMG Labtech). After recording of the baseline, 50µl freshly prepared oxaloacetic acid (10mM) was added to the sample, and the change in absorbance was recorded for 3 minutes at 412nm and 25˚C. To calculate the oxalo-acetic acid-dependent Δabsorbance, the baseline Δabsorbance was subtracted from Δabsorbance recorded after oxaloacetic acid addition. Citrate synthase activity was calculated using the same equation as for determining complex I activity, where the extinction co-efficient = 13.6 (TNB), volume = 1 and sample protein concentration = 0.005.

## ATP luciferase assay

20 fly heads were used for each replicate. To obtain fly heads, flies were placed into a 15ml falcon tube and snap-frozen in liquid nitrogen, before being vortexed 3 times for 5 seconds each. The tube contents were then tipped onto paper over dry ice, and heads were collected into an Eppendorf using a cold paintbrush. Snap-frozen fly heads were mashed using a pestle in 200µl buffer consisting of 100mM Tris (pH 7.4) and 4mM EDTA (pH 8.0) in ddH$_2$O. Ground fly

heads were immediately heated at 95˚C for 3 minutes and spun down at 14000g for 5 minutes. The resulting supernatant was then diluted either 1 in 50 or 1 in 2 in ddH2O. The ATP luciferase assay was performed using an ATP determination kit (A22066, Thermo Scientific), and following manufacturers protocols. Luminescence levels were read using a Clariostar monochromator microplate reader (BMG Labtech), and fly ATP levels were calculated from a standard curve produced using the ATP standards. Sample protein concentrations were determined using a Pierce BCA assay (Thermo Fisher Scientific), following manufacturer's protocols.

## Immunofluorescence and imaging

Images were taken using a Nikon A1R confocal microscope or a Nikon Vt-iSIM super resolution microscope with NIS Elements software.

Antibody fluorescence levels were quantified using ImageJ. For experiments measuring larval ATF4 or eIF2α-P expression levels, the cellular membrane marker CD8-GFP was used to identify cells to be quantified. Maximum intensity projections were then produced from Z-stacks of the dorsal side of the VNC. For quantification of ATF4 levels, 20 nuclei were identified in the green channel (based off of CD8-GFP expression), and circled using the 'freehand selections' tool. The mean fluorescence intensity of the nuclei selected was then recorded in the ATF4 expression channel (red channel). The same method was used to quantify eIF2 α-P levels, except cytoplasmic regions were circled instead of nuclei. Förster resonance energy transfer (FRET)–based ATP biosensor AT-NL [28] imaging and quantification was performed as described previously [29]. Primary antibodies were rat anti-ATF4 (1:200, [9]), were rabbit anti-P-eIF2α (1:500, anti-Phospho-eIF2α [Ser51], Cell Signaling Technology 9721). Secondary antibodies were goat anti-rabbit Alexa Fluor 546 (Invitrogen A11035), goat anti-rat Alexa Fluor 555 (Thermo Fisher A21434).

The number of SPLICS GFP puncta was quantified using NIS-Elements advanced research software (v5.01.00, Nikon). Images of larval VNCs were first cropped in the XY plane so that only whole neuromeres remained. Next, images were processed using the 'General Analysis (GA3)' tool. Within GA3, an analysis flow chart was set up to identify and quantify the number of bright 3D spots within the GFP channel. The same diameter, z-axis elongation and intensity thresholds were applied to all images. The number of GFP spots in the image was used as a measure of the number of contacts per neuromere.

A Vt-iSIM super resolution microscope (Nikon) was used to capture mitoGFP expression. Samples were first located on the slide using 20x magnification. Images were taken at 100x magnification and 1.5x zoom using an SR Apo TIRF 100x oil immersion lens (NA = 1.49). Z-stacks each 100nm in depth were taken of the dorsal side of the VNC. All Images taken were 1000 x 1440 pixels in size at a resolution of 43.29nm/pixel. Images were deconvolved within NIS-Elements software using a blind algorithm with 10 iterations. To quantify mitochondrial number and volume images were analysed using the Measurement tool in Volocity (v6.3.1, PerkinElmer). A region of interest (ROI) 1.6x1.6μM in height and width was placed within a cell, positioned so that in all Z-stacks, the ROI remained within the same cell and excluded the nucleus. For each condition, a total of 30 ROIs were analysed from across 6 images, with each ROI placed in a different cell.

## Western blot analysis

For each biological replicate, 10 male and 10 female flies were used. Flies were first homogenised in 40μl 1X sample buffer using a pestle in a 1.5ml Eppendorf and spun down at 14000g for 5 minutes. The 1X sample buffer consisted of 50mM Tris-HCl pH 6.8, 10% v/v glycerol, 2%

w/v SDS and 0.01% w/v bromophenol blue. 100mM dithiothreitol (DTT) was added and samples were boiled at 95°C for 5 minutes. Samples were stored at -20°C. Antibodies used were rabbit anti-VDAC (1:1000; ab14374, Abcam), mouse anti-Ndufs3 (1:500; ab14711, Abcam), rabbit anti-actin (1:5000; 4967, Cell Signalling Technology), anti-rabbit phospho-AMPK (Thr172; #2535, Cell Signalling Technology). Total protein staining was performed using the REVERT Total Protein Stain Kit (926–11010, LiCor).

## RNA sequencing (RNA-Seq) transcriptomic analysis

20 snap frozen fly heads (10 male and 10 female, 2 days old) were used for each replicate and placed into 100 µL of lysis buffer + β-mercaptoethanol from the Absolutely RNA Microprep kit (Agilent Technologies). Each genotype was prepared in quadruplicate. RNA was extracted from using the Absolutely RNA Microprep kit according to the manufacturer's protocol. The samples were sent on dry ice to Novogene Ltd. Sequencing libraries were generated using NEBNext Ultra TM RNA Library Prep Kit for Illumina (NEB, USA) following manufacturer's recommendations. The clustering of the index-coded samples was performed on a cBot Cluster Generation System using PE Cluster Kit cBot-HS (Illumina) according to the manufacturer's instructions. After cluster generation, the library preparations were sequenced on an Illumina platform and paired-end reads were generated. Raw data (raw reads) of FASTQ format were firstly processed through fastp. Paired-end clean reads were mapped to the *Drosophila*_melanogaster ensemble 102 genome using HISAT2 software. Featurecounts was used to count the read numbers mapped of each gene, then RPKM (Reads Per Kilobase of exon model per Million mapped reads) of each gene was calculated based on the length of the gene and reads count mapped to this gene [70]. Differential expression analysis between conditions was performed using DESeq2 R package [71]. P values were adjusted using the Benjamini and Hochberg's approach for controlling the False Discovery Rate (FDR). Genes with an adjusted P value < 0.05 found by DESeq2 were assigned as differentially expressed. GO enrichment analysis of differentially expressed genes was implemented by the clusterProfiler R package [72], in which gene length bias was corrected. GO terms with corrected P value less than 0.05 were considered significantly enriched by differential expressed genes. The clusterProfiler R package was used to test the statistical enrichment of differential expression genes in KEGG pathways. The heatmap was generated using SRPlot (http://www.bioinformatics.com.cn/srplot). *Drosophila* homologs of mouse ATF4 target genes [42] were identified using the Drosophila RNAi Screening Center Integrative Ortholog Prediction Tool [73] and only homologs with a high rank were included. RNA-Seq data have been deposited in NCBI's Gene Expression Omnibus [74] and are accessible through GEO Series accession number GSE214562 (https://www.ncbi.nlm.nih.gov/geo/query/acc.cgi?acc= GSE214562).

## Metabolomic analysis

Twenty adult flies (10 males and 10 females, 2 days old) for each replicate were snap-frozen on liquid nitrogen in a 15 ml Falcon tube and then vortexed for 5 s five times to decapitate. Heads were then quickly separated and stored at −80°C. Soluble metabolites were extracted directly from snap-frozen samples using cold acetonitrile/water (80/20, vol/vol) at 1 µl per 50 µg of samples. Fly heads were disrupted for 15 s by ultrasonication (Branson Sonifier 250) followed by vortexing for 3 times (60 s / time, 1 min on ice in between each vortex). Homogenized samples were incubated at −80°C overnight to precipitate proteins and subsequently thawed, vortexed, and centrifuged at 18,000 × g for 30 min at 4°C to pellet the debris. The supernatants were transferred to new vials and analysed by HPLC and high-resolution mass spectrometry and tandem mass spectrometry (MS/MS) as described in [9]. Raw values were normalised to

total ion count. Metabolomic data were analysed using MetaboAnalyst 5.0 [75]. Metabolites which were not detected in three or more replicates for at least one condition were excluded from analysis. For pairwise global metabolite analysis, individual metabolite concentrations were compared using multiple unpaired two-way t-tests, and p values were adjusted for the FDR using the Benjamini–Hochberg method for multiple-hypothesis testing [76]. FDR-adjusted p values <0.05 were considered statistically significant. Pairwise MSEA was performed using the 'quantitative enrichment analysis (QEA)' algorithm [77]. The overall metabolite profile (S4 Table) was used as the reference library. FDR-adjusted p-values≤0.05 were considered statistically significant. Using QEA, a p value and a Q statistic, a description of the correlation between compound concentration profiles and clinical outcomes, were generated for each metabolic pathway. The actual Q-statistic calculated for each pathway was divided by the expected Q-statistic to provide the 'enrichment ratio'. FDR-adjusted p-values≤0.05 were considered statistically significant.

## Transmission electron microscopy (TEM)

TEM of larval CNS tissue was performed as described previously [50]. Briefly, tissues from third instar wandering larvae were dissected and fixed in 0.1 M NaPO4, pH 7.4, 1% glutaraldehyde, and 4% formaldehyde, pH 7.3, overnight. Fixed larval preparations were washed 3× in 0.1 M $NaPO_4$ before incubation in $OsO_4$ (1% in 0.1 M $NaPO_4$, 1% potassium ferrocyanide (w/v) in 0.1 M $NaPO_4$; 2 h), dehydrated in an ethanol series, and embedded using Epon araldite resin. Sections were imaged using a transmission electron microscope (TECNAI 12 G2; FEI) with a camera (Soft Imaging Solutions MegaView; Olympus) and Tecnai user interface v2.1.8 and analySIS v3.2 (Soft Imaging Systems).

Two day old dissected adult fly brain samples were post-fixed with 1% (w/v) osmium tetroxide and 1% potassium ferrocyanide (w/v) in 0.1M sodium phosphate buffer (pH 7.4) for 1 hour before being washed and dehydrated through a graded acetone series. Samples were then infiltrated with increasing concentrations of SPURR epoxy resin/acetone mixture before being placed into 100% resin overnight with rotation. The following day, the samples were infiltrated further before embedding (with the dorsal face orientated toward the sectioning plane) and polymerised at 70˚C for 24 hours.

Ultrathin sections (50-70nm) were prepared using a Leica UC7 ultramicrotome (Leica microsystems, Vienna), mounted on grids and contrasted using UranyLess (22409 Electron Microscopy Sciences, USA) and lead citrate (22410 Electron Microscopy Sciences, USA). Samples were examined on a JEOL JEM 1400 Flash (JEOL, Japan) transmission microscope operated at 80 kV and images were acquired with a JEOL Flash Camera.

Mitochondrial contacts were quantified in ImageJ (version 1.52). Mitochondria with ER contacts were identified by observing a 30nm or less distance between the mitochondria and ER. For positive contacts, mitochondria circumferences were measured and recorded. The length of the ER with a distance <30nm was also measured and then divided by the mitochondria circumference to calculate the proportion of the mitochondrial membrane in contact with the ER.

## Statistical analyses

Continuous data are expressed as mean ± S.E.M unless stated otherwise. Non-continuous data are expressed as percentages unless stated otherwise. Excluding metabolomic data, all data were analysed using Prism 8 (GraphPad). Student's unpaired two-way t-tests were used for pairwise comparisons of continuous data. An F-test was used to test for unequal variances, and where significant, Welch's correction was applied to the t-test. A one-way ANOVA with

Tukey's post-hoc test was used for continuous data with multiple comparisons. Where variances were unequal, Welch's ANOVA followed by Dunnett's T3 multiple comparisons test was used. Chi-squared test and Fisher's tests were used for non-continuous data, and were applied to the raw values rather than percentages. The log-rank test was used for lifespan data. For the NADH dehydrogenase activity assay data were normalised to the control and then transformed using log base 2. P values $<0.05$ were considered significant; * $p<0.05$, ** $p<0.01$, *** $p<0.001$, ****$p<0.0001$, n.s. non-significant. Experimental conditions where all values were equal to 0 were not included in statistical analysis.

## Supporting information

**S1 Fig. Validation of ND-75**$^{KD}$**.** (A) qRT-PCR analysis of ND-75 mRNA levels from adult flies with ubiqitious ND-75$^{KD}$ using *Da-GS-Gal4*. n = 8 biological replicates for all genotypes. (B) Complex I activity in mitochondria isolated from adult flies with ubiqitious ND-75$^{KD}$ using *Da-GS-Gal4*. Control n = 5, ND-75$^{KD}$ n = 5 biological replicates. (C) Western blot analysis of ND-30 expression in adult flies with ubiqitious ND-75$^{KD}$ using *Da-GS-Gal4*. (D, E) Quantification of ND-30 expression relative to actin (D) and the mitochondrial outer membrane protein VDAC (E). Control n = 3, ND-75$^{KD}$ n = 3 biological replicates. (F) Heterozygous ND-75 RNAi flies do not have a climbing phenotype. n = 11 for all genotypes. (H) (G) Ubiqitious ND-75$^{KD}$ using *Da-GS-Gal4* causes a strong climbing phenotype. Control n = 11, ND-75$^{KD}$ n = 11. (H) Reduced climbing abilty of flies expressing an alternative ND-75 shRNA (HMS00854) in neurons with *Tub-Gal80$^{ts}$*; *nSyb-Gal4*. Control n = 11, ND-75$^{KD}$ n = 11. Controls are *Da-GS-Gal4* or *Tub-Gal80$^{ts}$*; *nSyb-Gal4* hemizygotes. 5 day old male and female flies were used in (A)-(E). 1 day old male flies were used in (F, H). 5 day old male flies were used in (G). Data are represented as mean ± SEM and were analysed using Student's unpaired t-test. ns not significant, *p<0.05, ***p < 0.001, ****p < 0.0001.
(TIF)

**S2 Fig. ND-75**$^{KD}$ **does not alter ATP levels in larval neurons.** (A-B") Fluorescence emission of a control ATP insensitive probe (AT-RK) (A-A") and the ATP-sensing AT-NL probe (B-B") when excited at 405nm (CFP, cyan) and 488nm (FRET, yellow) in larval motor neurons using *OK371-Gal4*. (C) Quantification of CFP/FRET fluorescence. Control n = 5, ND-75$^{KD}$ n = 5 larval CNS. (D-E") AT-NL fluorescence emission of control and ND-75$^{KD}$ larval motor neurons when excited at 405nm (CFP, cyan) and 488nm (FRET, yellow) using *OK371-Gal4*. AT-RK n = 9, AT-NL n = 11 larval CNS. (F) Quantification of CFP/FRET fluorescence in control and ND-75$^{KD}$ neurons. Control n = 11, ND-75$^{KD}$ n = 11 larval CNS. (G) ATP levels measured in whole heads from control flies or with with pan-neuronal ND-75$^{KD}$ using *Tub-Gal80$^{ts}$*; *nSyb-Gal4*. Control n = 8, ND-75$^{KD}$ n = 8 biological replicates. (H)Lactate levels from metabolomic analysis of pan-neuronal ND-75$^{KD}$ using *Tub-Gal80$^{ts}$*; *nSyb-Gal4*. Control n = 6, ND-75$^{KD}$ n = 6.. 2 day old male and female flies were used in (G) and (H). Controls are *OK371-Gal4* or *Tub-Gal80$^{ts}$*; *nSyb-Gal4* hemizygotes. Student's t test. Data are represented as mean ± SEM. ns not significant, *p < 0.05.
(TIF)

**S3 Fig. Validation of the SPLICS reporter.** (A, B) Visualisation of ER-mitochondria contacts by SPLICS expression in larval motor neurons using *OK371-Gal4* in control (A) and with IP$_3$R knockdown (B). (C) Quantification of ER-mitochondria contacts. Control n = 6, IP$_3$R n = 6 larval CNS. (D) Climbing ability of flies with Hsc70-3$^{[K97S]}$ and Hsc70-3$^{[D231S]}$ overexpression in neurons using *nSyb-Gal4*. Control n = 15, Hsc70-3$^{K97S}$ n = 15. Controls are *OK371-Gal4* or *nSyb*-Gal4 hemizygotes. Data are represented as mean ± SEM and were analysed using

Student's t test. ***p < 0.001, ****p < 0.0001 (E, F) GO cellular component (E) and molecular function (F) analyses of genes with significantly increased expression in head tissue from pan-neuronal ND-75$^{KD}$ using *Tub-Gal80$^{ts}$*; *nSyb*-Gal4.
(TIF)

**S4 Fig. Altered ROS levels in ND-75$^{KD}$ larval neurons do not contribute to neuronal dysfunction.** (A-D) mito-roGFP2-Grx1 expression showing merge of 405nm excitation (blue) and 488nm excitation (green) emission fluorescence in control (A) and ND-75$^{KD}$ (B), NDI1 expression (C) and or ND-75$^{KD}$ with NDI1(D) larval motor neurons using *OK371-Gal4*. (E) Quantification of mito-roGFP2-Grx1 405nm and 488nm excitation fluorescence. n = 15 larval CNS for all genotypes. (F-J) Climbing ability of pan-neuronal ND-75$^{KD}$, using *Tub-Gal80$^{ts}$*; *nSyb-Gal4*, with: (F) SOD1 RNAi, control n = 25, SOD1 RNAi n = 24, ND-75$^{KD}$ n = 24, ND-75$^{KD}$;SOD1 RNAi n = 22 flies. (G) SOD2 RNAi, control n = 21, SOD2 RNAi n = 21, ND-75$^{KD}$ n = 7, ND-75$^{KD}$;SOD2 RNAi n = 7 flies. (H) SOD2 overexpression (o/e), control n = 25, SOD2 o/e n = 25, ND-75$^{KD}$ n = 5, ND-75$^{KD}$;SOD2 o/e n = 5 flies. (I) catalase overexpression (o/e), control n = 27, catalase o/e n = 24, ND-75$^{KD}$ n = 11 ND-75$^{KD}$; catalase o/e n = 7 flies. (J) mito-catalase overexpression (o/e), control n = 15, mitocatalase o/e n = 15, ND-75$^{KD}$ n = 3 ND-75$^{KD}$; mitocatalase o/e n = 5 flies. Controls are *OK371-Gal4* or *Tub-Gal80$^{ts}$*; *nSyb-Gal4* hemizygotes. 1 day old male flies were used in (G-J). Student's t test. Data are represented as mean ± SEM. n.s. not significant, *p < 0.05, **p<0.01, ***p<0.001, ****p < 0.0001.
(TIF)

**S1 Table. Genes with significantly increased expression in head tissue from pan-neuronal ND-75$^{KD}$ using Tub-Gal80$^{ts}$; nSyb-Gal4.**
(XLSX)

**S2 Table. Genes with significantly decreased expression in head tissue from pan-neuronal ND-75$^{KD}$ using Tub-Gal80$^{ts}$; nSyb-Gal4.** In column N, complex I genes are highlighted in green, complex II genes highlighted in blue (also components of the TCA cycle), complex III genes highlighted in red, compex IV genes highlighted in yellow, complex V genes highlighted in gray. TCA cycle genes are highlighted in orange (except SdhA-D in blue).
(XLSX)

**S3 Table. Results of GO analysis showing significantly enriched biological process (BP), cellular component (CC) and molecular function (MF) in genes with significantly increased expression in ND-75$^{KD}$ flies.**
(XLSX)

**S4 Table. Results of GO analysis showing significantly enriched biological process (BP), cellular component (CC) and molecular function (MF) in genes with significantly decreased expression in ND-75$^{KD}$ flies.**
(XLSX)

**S5 Table. Drosophila homologs of mammalian ATF4 target genes with significantly increased expression in ND-75$^{KD}$ flies.**
(XLSX)

**S6 Table. Metabolite levels, normalised to total ion count, from control (hemizygous Tub-Gal80$^{ts}$; nSyb-Gal4), or pan-neuronal ND-75$^{KD}$, NDI1, or ND-75$^{KD}$ + NDI1 using Tub-Gal80$^{ts}$; nSyb-Gal4 fly head tissue.** Colour of metabolite ID stands for: black = identity confirmed by standard or MS2, red = cannot separate, orange = identity not confirmed.
(XLSX)

**S7 Table. Metabolites with significantly altered levels in head tissue from control versus pan-neuronal ND-75$^{KD}$ using Tub-Gal80$^{ts}$; nSyb-Gal4.**
(XLSX)

**S8 Table. Details of Drosophila strains used.**
(DOCX)

## Acknowledgments

We are grateful to Rita Sousa-Nunes, Nazif Alic, Alex Whitworth and Darren Williams for fly stocks and the KCL Centre for Ultrastructural Imaging for technical assistance. Stocks obtained from the Bloomington *Drosophila* Stock Center (NIH P40OD018537) and the Vienna *Drosophila* Resource Center and FlyORF were used in this study. We are grateful to Ben Kottler for assistance with behavioural analyses, Sabrina Liu for help with climbing assays, Laura Raik for help with the feeding assay, Clare Steele-King of the York Imaging and Cytometry Facility for help with TEM, Peng Gao and the Robert H. Lurie Cancer Center Metabolomics Core at Northwestern University Feinberg School of Medicine for metabolomics analysis. We thank the Wohl Cellular Imaging Centre at King's College London for help with light microscopy.

## Author Contributions

**Conceptualization:** Joseph M. Bateman.

**Data curation:** Emma L. Hamer.

**Formal analysis:** Lucy Granat, Debbra Y. Knorr, Daniel C. Ranson, Emma L. Hamer, Ram Prosad Chakrabarty, Francesca Mattedi, Laura Fort-Aznar, Joseph M. Bateman.

**Funding acquisition:** Sean T. Sweeney, Navdeep S. Chandel, Joseph M. Bateman.

**Investigation:** Joseph M. Bateman.

**Methodology:** Navdeep S. Chandel.

**Resources:** Francesca Mattedi, Frank Hirth, Navdeep S. Chandel.

**Supervision:** Sean T. Sweeney, Alessio Vagnoni, Navdeep S. Chandel, Joseph M. Bateman.

**Writing – original draft:** Joseph M. Bateman.

**Writing – review & editing:** Lucy Granat, Navdeep S. Chandel, Joseph M. Bateman.

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
