## [Decision Letter · Decision Letter 0]

18 Dec 2022

Dear Dr Bateman,

Thank you very much for submitting your Research Article entitled 'Yeast NDI1 reconfigures neuronal metabolism and prevents the unfolded protein response in mitochondrial complex I deficiency' to PLOS Genetics.

The manuscript was fully evaluated at the editorial level and by independent peer reviewers. The reviewers appreciated the attention to an important problem, but raised some substantial concerns about the current manuscript. Based on the reviews, we will not be able to accept this version of the manuscript, but we would be willing to review a much-revised version. We cannot, of course, promise publication at that time.

If you decide to revise the manuscript for further consideration at PLOS Genetics, please aim to resubmit within the next 60 days, unless it will take extra time to address the concerns of the reviewers, in which case we would appreciate an expected resubmission date by email to plosgenetics@plos.org.

We are sorry that we cannot be more positive about your manuscript at this stage. Please do not hesitate to contact us if you have any concerns or questions.

Yours sincerely,

Aleksandra Trifunovic

Academic Editor

PLOS Genetics

Gregory P. Copenhaver

Editor-in-Chief

PLOS Genetics

Reviewer's Responses to Questions

**Comments to the Authors:**

Reviewer #1: Mitochondrial complex I deficiency causes severe neurological diseases including Leigh syndrome. The pathogenic mechanism of the disease is poorly understood. In this manuscript, the authors modelled complex I deficiency in Drosophila by knocking down the mitochondrial complex I subunit ND-75 (NDUFS1) specifically in neurons. They found that complex I deficiency affects neuronal function and lifespan, without reducing the overall cellular ATP levels. In the mutant flies, they observed increased mitochondrial density, reduced endoplasmic reticulum-mitochondria contacts and activation of the endoplasmic reticulum unfolded protein response (UPR) in neurons. The data also suggested changes to mitochondrial metabolism. Interestingly, expression of the yeast NADH dehydrogenase NDI1, which does not pump protons, corrected some of these metabolic changes, reinstated endoplasmic reticulum-mitochondria contacts, prevented UPR activation and rescued the behavioral and lifespan phenotypes caused by complex I deficiency. Based on these data, the authors concluded that aberrant NAD+/NADH ratio and metabolic disruption in the complex I-deficient neurons drive UPR activation and neuronal pathology.

This is a well executed study. Most of the data (behavioral and biochemical) presented are strong. Particularly, the behavioral and survival defects and altered stress signaling caused by complex I deficiency can be robustly corrected by the yeast NDI1 gene. This suggests that metabolic defect other than energy deficiency may contribute to the loss of neuronal function. Although the exact mechanism remains unidentified, this finding may open a new direction for the better understanding of pathologies caused by complex I-deficiency. The following recommendations may improve the manuscript.

Major recommendations:

1) Given that mitochondria and ER are under stress in complex I-deficient neurons, the data from the split-GFP experiment for evaluating the mito-ER contact sites may be at risk of overinterpretation. Under these conditions, there is no guaranty that the split GFP constructs are efficiently targeted to the organelles. This may confound the interpretation of the data. This part of the data is not critical for the central theme of the manuscript. The authors should consider removing it from the manuscript.

2) ATF4 activation and phosphorylation of eIF2alpha are increased in ND-75KD flies. The authors assume that the (classic ER) UPR is activated. How can this be distinguished from ISR activation that is known to be stimulated by mitochondrial dysfunction? In this context, although the dominant negative alleles of Hsc70-3 cause pupal lethality and climbing defects, there is no guaranty that this entails the same physiological consequences as complex I deficiency. As such, the conclusion that “This finding strongly suggests that activation of the UPR contributes to neuronal dysfunction in ND-75KD flies” is weak. Accordingly, the authors may consider reword the title.

3) It looks that mitochondrial density is increased in ND-75KD flies. There are no data supporting morphological changes to mitochondria. The authors need to reword the text. On the other hand, increased mitochondrial density is not consistent with the overall repression of nuclear genes encoding mitochondrial proteins. Any explanation for this discrepancy?

4) To exclude the possibility that the ectopic expression of NDI1 induces compensatory adaptation to mitochondrial damage independent of NADH dehydrogenase activity, it would be important to show the levels of NADH dehydrogenase in the rescued flies. Either positive or negative data would be helpful for better understanding how neuronal function is affected.

Minor points:

1) The “drastic” changes in the expression of genes encoding the ETC and TCA cycle are embedded in the supplemental tables that are not easy to navigate by the readers. It would be good if the authors can make a figure illustrating the top hits and show it in one of the figures.

2) Page 4, paragraph 2 – “Importantly, targeting stress signaling pathways, including the UPR and ISR, has shown therapeutic potential in animal models”. It would be good if the authors can cite the relevant references.

3) The asterisks denoting the statistical values in the figures throughout the manuscript need to be annotated differently as they are all too small and sometime difficulty to differentiate from the data points.

4) Page 6, paragraph 3 – delete “have”.

5) Page 36; Figure S2 – (H) panel is missing.

Reviewer #2: CI deficiency leads to severe neuropathologies and childhood mortality. In this study, Granat et al model CI deficiency in Drosophila by neuron-specific RNAi of the 75kDa subunit (NDUFS1). The authors successfully exploit the benefits of Drosophila as an in vivo system, and show convincingly that the KD flies exhibit severe neurological dysfunction (climbing, locomotion, seizures), which impacts on their survival, and provide evidence implicating the UPR in this process. Overall, this is an interesting and well written manuscript, which will have broad appeal and relevance to many fields (mitochondria, neurodegeneration, metabolism, fly models of genetic disease).

While the data presented are clear and compelling, there are several points that would benefit from being addressed or discussed prior to publication.

1) Control lines - Throughout the manuscript, ‘control’ refers to the driver line alone. For selected experiments, it would be helpful to have the additional UAS-RNAi alone control (e.g. q-RT-PCR in Fig S1A). This would confirm that the transgene is not leaky i.e. expressed in the absence of the driver, which would be important for conclusions relating to the tissue-selectivity of subsequent manipulations.

Similarly, there may be effects due to differences in genetic backgrounds between experimental groups. The TRIP collection includes several lines that could be used as more appropriate background controls (https://fgr.hms.harvard.edu/trip-rnai-control-fly-stocks).

Relating to Figs 4-6, it would be valuable to confirm if the level of ND-75 knock-down is the same when the UAS-RNAi line is combined with the additional UAS-Ndi1 line.

2) Mitochondrial morphology - Morphological changes to mitochondria are evident from the mitoGFP imaging, but based on Figs 2A-B and given their network-like nature, it is not entirely clear how a ‘number of mitochondria’ was reached. Would area/volume be more appropriate?

3) Experimental samples (sex, age, temperature) - Regarding the samples used for analysis, many of the experiments appear to pool male and female flies. The rationale for this mixing is unclear, since many metabolic, physiological and behavioural traits (e.g. feeding, activity, survival) have been widely shown to be sex-dependent in flies. Therefore, pooling may introduce biological variability or mask potentially interesting sex-specific effects? For the climbing assays, it is not clear if this was done on male and/or female flies? For the lifespans, which were done only in males, do the authors observe a similar decrease in survival of female flies?

Given the severe shortening of lifespan in the ND-75 KD flies (median of 5 days for males), it is understandable that experiments need to be performed on very young flies. However, the age of the flies used is not entirely clear. For example, the RNA-seq is described as being performed on 2 day old flies. Whereas for some experiments, flies are described in the methods as being used the day after eclosion. Are these the same age i.e. do the authors consider the day of eclosion as day 0 or day 1? Since flies are still in the process of maturing during this early period (e.g. the gut, which may potentially impact some of their metabolic readouts), it would be important to know the age of samples. This would help in terms of comparison with other studies and future data reproducibility.

For the Geneswitch experiments, it was not clear why the flies were incubated for 5 days at 29ºC (since this is not a temperature-sensitive driver)? It would be helpful to explain the rationale, as all other adult experiments appear to have been performed at 25ºC.

Overall, the exact type of sample (sex & age) should be fully defined in the figure and/or legend, not just in the methods section, which would help facilitate data interpretation.

4) Fly strains - Regarding the fly strains used in the study, in addition to the simplified stock name, please state the full genotype including genetic background for each line (Table S5). Also, it would be helpful to include original publication references for any lines that are not available in stock centres. Regarding the ND-75 RNAi lines, please mention more explicitly in the manuscript that these are from the TRIP collection (PMID 26320097).

5) Protein quantification - The VDAC band intensity in Fig S1C appears to be highly variable between samples, which complicates the relative quantification in Fig S1E. Could the authors speculate on this? Regarding CI quantification, the data indicate loss of the whole complex. Have the authors considered potential changes to other complexes in the mitochondrial ETC?

6) A previous study has shown that ubiquitous RNAi of CI subunits can extend fly lifespan (PMID 19747824). Could the authors comment on how their neuronal-specific data relate to these findings? Did the authors try lifespans for their ND-75 KD with the inducible ubiquitous daGS driver?

Reviewer #3: The manuscript by Granat et al. describes the effects of complex I subunit ND-75 (NDUFS1) knockdown in Drosophila neurons. ND-75 is a core and the largest subunit of complex I, and its loss resulted in severe motor dysfunction, neurological phenotypes, and reduced lifespan. Molecularly ND-75 knockdown in neurons caused changes in cellular metabolism, transcriptional response, ER-mitochondria contact sites, mitochondrial morphology and UPR. The authors aimed to rescue these phenotypical and molecular phenotypes by expressing the yeast NADH dehydrogenase NDI1 in neurons, which led to almost a complete reversal of the observed parameters.

Overall, the findings are interesting and highlight the importance of NADH dehydrogenase activity of complex I in neurological mitochondrial dysfunction. It is a nice addition to the growing number of studies linking mitochondrial dysfunction to the ISR in a mitochondrial defect- and metabolic state-specific manner.

The authors utilized a variety of imaging techniques and multi-omics approaches. Moreover, the manuscript is clearly written and easy to follow. There are, however, some loose ends that needs tightening. A number of comments and suggestions to improve the study include:

1) ND-75KD flies show an extreme lifespan reduction, with a median survival of 5 days. Hegde et al.’s global and neuronal knockdowns of ND-75 in flies, on the other hand, resulted in median lifespans of 27 and 25 days, respectively. These differences in survival and these models in general should be discussed.

2) Some mitochondrial dysfunction fly models have been shown to have severe feeding difficulties (i.e., dNDUFS4-knockdown flies). Considering the severe neurological phenotypes of the ND-75KD flies, have the authors checked food intake or observed any feeding difficulties in the animals? Starvation could also increase various stress pathways.

3) The results presented in Figure S2 and related to ATP levels are somewhat surprising considering the complex I decrease. For example, dND2 mutant flies, like the current ND-75KD flies, display many of the hallmarks of mitochondrial diseases, including reduced lifespan and signs of neurodegeneration. However, dND2 flies have lower levels of ATP.

The ATP levels in the adult heads of ND-75KD flies show a tendency towards a decrease (Figure S2G). The sample size is not indicated in the figure legends, but this experiment might benefit from increasing the sample size. One would argue that increased glycolysis might also account for the unchanged ATP levels. In that case, this claim should be supported by experimental evidence.

Similarly, in Figures S2G-I, the p-AMPK levels in the adult heads signal to a possible increase. The blot should definitely include total AMPK levels and quantification of p-AMPK/AMPK ratio.

4) The transcriptomic analysis of head tissue from ND-75KD flies revealed a battery of changed transcripts. Considering the mitochondrial defect and increased ER stress in this model, it would be very useful to further investigate the data in light of classical mitochondrial stress responses, i.e., mitochondrial unfolded protein response, antioxidant response, components of mitochondrial integrated stress response, autophagy/mitophagy, etc. Additional bioinformatic analyses concentrating on mitochondrial stress responses would be very beneficial.

5) Complex I is one of the major reactive oxygen species (ROS) producers in mitochondria (ROS is being used as an umbrella term for superoxide and hydrogen peroxide in this context). NDI1 has also been shown to increase reverse electron transport-mediated ROS (RET-ROS) levels and regulate stress adaptation, lifespan and even mitochondrial function in Drosophila.

As the ND-75KD flies have huge decrease in complex I activity and increased stress responses, measuring ROS levels and some players of antioxidant defenses in the brain before and after NDI1 introduction could add great value to the study.

6) The animal models of mitochondrial dysfunction generally present with increased lactate levels. In this case, lactate elevation was not detected in the metabolomics experiment. This point can be discussed.

Stylistic comments:

1) In general, the experimental results of larval and adult knockdowns are hard to distinguish just looking at the figures. It would be useful to have some indication for this separation.

2) The sample sizes must be added to the figure legends, where applicable.

3) Line numbers should be added to the manuscript as required by the journal.

4) The references should be styled according to the journal submission guidelines.

5) “… a readout of AMP/ATP ratio (Supplemental Fig. S2G-F).” should be Supplemental Fig. S2G-I.

6) In the Supplementary Table 1, some gene names have been changed into dates in Excel. These should be corrected.

**Have all data underlying the figures and results presented in the manuscript been provided?**

Reviewer #1: Yes

Reviewer #2: Yes

Reviewer #3: Yes

PLOS authors have the option to publish the peer review history of their article (what does this mean?). If published, this will include your full peer review and any attached files.

Reviewer #1: **Yes: **Xin Jie Chen

Reviewer #2: No

Reviewer #3: No

---

## [Decision Letter · Decision Letter 1]

22 May 2023

Dear Dr Bateman,

We are pleased to inform you that your manuscript entitled "Yeast NDI1 reconfigures neuronal metabolism and prevents the unfolded protein response in mitochondrial complex I deficiency" has been editorially accepted for publication in PLOS Genetics. Congratulations!

Please note the minor note by Reviewer #3 (below) regarding the numbering of the supplemental figures which you can attend to as you prepare your final draft for the production team (the editorial team will not need to re-evaluate). 

Yours sincerely,

Aleksandra Trifunovic

Academic Editor

PLOS Genetics

Gregory P. Copenhaver

Editor-in-Chief

PLOS Genetics

Comments from the reviewers (if applicable):

Reviewer's Responses to Questions

**Comments to the Authors:**

Reviewer #1: The manuscript should be ready for publication. There is one typo on page 11, line 15 ("underly") that can be corrected at the proofreading stage.

Reviewer #2: The authors have satisfactorily addressed the comments raised at peer review. Overall the revised manuscript is improved, the amendments are clearly outlined in their rebuttal and within the corrected paper, and I have no further comments.

Reviewer #3: The authors have satisfactorily responded to my comments and suggestions, and improved the manuscript. I would like to thank the authors for their hard work.

One minor point: Supplemental figures are mislabeled. According to their use in the text, Figure S4 should be S2; S2 should be S3; S3 should be S2.

**Have all data underlying the figures and results presented in the manuscript been provided?**

Reviewer #1: Yes

Reviewer #2: Yes

Reviewer #3: Yes

PLOS authors have the option to publish the peer review history of their article (what does this mean?). If published, this will include your full peer review and any attached files.

Reviewer #1: **Yes: **Xin Jie Chen

Reviewer #2: No

Reviewer #3: No

**Data Deposition**

http://datadryad.org/submit?journalID=pgenetics&manu=PGENETICS-D-22-01234R1

**Press Queries**

---

## [Editor Report · Acceptance letter]

29 Jun 2023

PGENETICS-D-22-01234R1 

Yeast NDI1 reconfigures neuronal metabolism and prevents the unfolded protein response in mitochondrial complex I deficiency 

Dear Dr Bateman, 

We are pleased to inform you that your manuscript entitled "Yeast NDI1 reconfigures neuronal metabolism and prevents the unfolded protein response in mitochondrial complex I deficiency" has been formally accepted for publication in PLOS Genetics! Your manuscript is now with our production department and you will be notified of the publication date in due course.

With kind regards,

Lilla Horvath

PLOS Genetics

On behalf of:
